# EXTENDING MYERSON'S OPTIMAL AUCTIONS TO CORRELATED BIDDERS VIA NEURAL NETWORK INTERPOLATION

## ABSTRACT

We aim to design revenue-maximizing single-item auctions that are deterministic, strategy-proof and ex post individually rational. Myerson's seminal work on optimal auction design solved this problem for independent bidders. Myerson introduced the novel concept of virtual valuation and showed that revenue maximization is equivalent to virtual valuation maximization. Coincidentally, by greedily allocating the item to the bidder with the highest (ironed) virtual valuation, the resulting allocation is guaranteed to be monotone – a necessary and sufficient condition for strategy-proofness.

For correlated bidders, Myerson's greedy allocation no longer guarantees monotonicity/strategy-proofness. We propose a simple yet empirically effective approach for designing near-optimal auctions for correlated bidders. We train a neural network to interpolate the greedy allocation, while enforcing that the interpolation must be verifiably monotone.

Empirically, our method consistently achieves near-optimal revenue across a wide range of distributions, including adversarially generated cases. Compared to existing baselines, our approach shows substantial improvement, often reducing the gap to the (unattainable) greedy upper bound by an order of magnitude.

Furthermore, we demonstrate the generality of our approach by extending it to multi-unit auctions with unit demand, where we achieve similarly strong performance. Additionally, our verification techniques can be integrated into the Regret-Net framework to design fully strategy-proof auctions.

## 1 INTRODUCTION

Myerson's landmark work on optimal auctions (Myerson, 1981) laid the foundation for the field of mechanism design, for which Myerson was awarded the 2017 Nobel Memorial Prize in Economic Sciences. Myerson (1981) solved the problem of revenue-maximizing single-item auction design for *independent* bidders. The derived optimal auction is *deterministic*, *strategy-proof* and *ex post individually rational*. Myerson's results on optimal auctions can be summarized as follows: 1) The allocation rule uniquely characterizes the payment rule. Given an allocation rule, there is only one way to charge the payments without violating either strategy-proofness or individual rationality. As a result, the mechanism design task comes down to designing only the allocation. 2) An allocation rule, or in other words, the "complete" mechanism it corresponds to, is strategy-proof if and only if a *monotonicity* condition holds. In the context of deterministic auctions, monotonicity means that the winner must still win if she raises her own bid while the other bids stay the same. Eventually, monotonicity is the only mechanism design constraint. 3) Myerson proposed the novel concept of *virtual valuation* and also a fairly intricate mathematical maneuver called *ironing*. Myerson showed that the actual bids can be converted to (ironed) virtual valuations and the *expected revenue* is mathematically identical to the *expected virtual valuation*. This leads to a simple greedy approach for maximizing revenue – just allocate the item to the bidder with the highest virtual valuation.[1] Coincidentally (in the sense that the mathematics underlying the model allows for such a simple characterization), My-

---

[1]The item is not allocated when all virtual valuations are below 0.

erson proved that the greedy allocation guarantees monotonicity with the independence assumption, implying that the greedy allocation is optimal for independent bidders.

For correlated bidders, a natural way to extend Myerson's optimal auction is to greedily allocate based on the *conditional* virtual valuations (*i.e.*, virtual valuations calculated using *conditional* distributions). Unfortunately, such extension often results in non-monotonic allocations (*i.e.*, not strategy-proof). We therefore need to design alternative allocation rules. Optimal allocation design for correlated bidders is actually hard – Papadimitriou & Pierrakos (2011) gave a reduction from 3CatSat, which is NP-hard to approximate better than 103/104, and Caragiannis et al. (2016) gave a reduction from MAX-NM 3SAT, leading to a bound of 63/64.

Unlike the above theoretical works, our paper focuses on the computational task of designing a near-optimal allocation rule *given a specific correlated distribution*. Our proposed approach is simple, yet empirically effective. We argue that having a simple (and effective!) solution is not a disadvantage, but rather an advantage. We summarize our contributions and main techniques as follows.

**Contribution 1: Neural network interpolation as a verification tool for strategy-proofness.**

Given a specific correlated distribution, a natural first step is to check whether Myerson's greedy allocation remains monotone *despite the correlation*. Note that it is infeasible to enumerate all bid profiles (*i.e.*, use "for loops") to verify monotonicity/strategy-proofness, which can only prove a negative and is not scalable. Under our approach, we supervise a neural network to mimic Myerson's greedy allocation, and then apply neural network verification techniques to check whether the *neural network interpolation version* of the greedy allocation remains monotone (strategy-proof) under the given correlated distribution.[2] Specifically, we use a multilayer perceptron (MLP) with ReLU activation to model the allocation function. This architecture allows us to *exactly verify* the monotonicity of a trained allocation using mixed-integer programming (MIP). In terms of training, for every training sample, which is a bid profile drawn from the given correlated distribution, we calculate the bidders' conditional virtual valuations and instruct the network to allocate to whoever has the highest virtual valuation (if the highest virtual valuation is at least 0).

It is worth noting that we encountered a somewhat surprising observation in experiments. For many correlated distributions, Myerson's greedy allocation is **not** monotone, but the supervision result is a verifiably monotone interpolation.[3] This can be explained by our tiny network size, limited by the MIP verification step. Tiny networks are not *capable* of learning the fine details of the greedy allocation, which turns out to be a silver lining, as the tiny networks are "glossing over" the monotonicity violations of the greedy allocation and only interpolate the macro trend. For the greedy allocation illustrated in Figure 1b, which contains monotonicity violations (i.e., the blue points enclosed by red and the red points enclosed by blue, of which we will provide proper definitions in Section 2), the learned allocation becomes Figure 1a at the very end, with the violations glossed over.[4]

We propose that *neural network interpolation can serve as a general tool for verifying mechanism properties*, including but not limited to strategy-proofness. The mechanism may be expressed mathematically or in code. By applying supervision to train a neural network to interpolate the mechanism, we can then apply neural network verification techniques to verify the mechanism properties.

**Contribution 2: A suite of monotonicity-seeking techniques, including counterexample-guided training and post-processing monotonicity fix, both enabled by the MLP+ReLU architecture.**

---

[2]Although there are neural network architectures that ensure "monotonicity", such as the *min-max networks* (Sill, 1997), it is important to note that the notion of "monotonicity" represented by min-max networks is different from "allocation monotonicity" in mechanism design. It is not clear how to use min-max networks to represent the full space of strategy-proof allocations. We defer the detailed discussion on this difference to Appendix A.9, where we also provide an example showing that the min-max network based MyersonNet (Dütting et al., 2019), when extended to correlated bidders via the *Lopomo assumptions* (Roughgarden & Talgam-Cohen, 2013), cannot express the optimal auction and leads to significant revenue loss.

[3]In our experiments, we found that while the greedy allocation is rarely monotone, the deviations are minimal. For 2-bidder cases where it is scalable to run the classical mixed-integer-programming approach to automated mechanism design (AMD) (Conitzer & Sandholm, 2002), we observe that greedy and AMD's allocation (monotone) generally coincide, differing in only a small percentage of bid profiles, more in A.6.1.

[4]We defer the full story behind Figure 1 to later sections. The learned allocation is actually Figure 1c, which needs to go through another *revenue fix* process (Section 3) before arriving at the allocation in Figure 1a.

Besides verification, the MLP+ReLU architecture enables two monotonicity-seeking techniques. First, when verification fails, the "by-products" of the verification MIP are bid profiles that violate monotonicity, which can be used in *counterexample-guided training* (Sivaraman et al., 2020). The idea is to punish the violation in follow-up training until the violation disappears. Second, if counterexample-guided training still fails to reach monotonicity, then we can implement a post-processing *monotonicity fix* to ensure strategy-proofness.[5] We construct another MIP based on the network parameters, which decides how much to push up the winner's offer while fixing the other bids. The new offer ensures that the winner remains the winner regardless of any bid increment.

As mentioned earlier, limited by the MIP-based verification step, we are restricted to tiny networks. As described by the Lottery Ticket Hypothesis (Frankle & Carbin, 2019), for tiny networks, having a "lucky" initialization is important. Motivated by this, the last monotonicity-seeking technique is simply *repeated trials*, which turns out to be effective for most distributions that we experimented on. That is, when we encounter either a poorly performing allocation or when monotonicity verification fails, we simply train again from scratch with a fresh initialization, until a near-optimal and verifiably monotone interpolation appears. In one case study on a specific distribution (Figure 3), we show that counterexample-guided training and the post-processing monotonicity fix only bring *statistically insignificant* revenue improvement compared to repeated trials. That is, for this distribution, *repeated trials are all we need*. On the other hand, there are indeed distributions under which repeated trials do not work, and we do need the full suite of techniques, including counterexample-guided training and the post-processing monotonicity fix, to achieve near-optimal revenue. In A.6.2, we generate 2 adversarial distributions with the help of a separate mixed-integer-programming heuristic. In A.7, we conducted 100 training trials for each of these distributions, corresponding to *hundreds of hours* of computation (mostly on solving MIPs). While we do manage to get verifiably monotone allocations via (now a lot more) repeated trials, the achieved revenue is much worse than the revenue obtained using the full suite of techniques, which is near-optimal. That is, for the two adversarial distributions, even by trying for hundreds of hours, we still cannot reach a near-optimal auction. This demonstrates the need for counterexample-guided training and the monotonicity fix at least in some cases. Lastly, the monotonicity fix offers the peace of mind that regardless of the distribution, our approach can always deliver strategy-proofness.

**Contribution 3: Extensive experiments demonstrating near-optimal revenue across all distributions tested, with substantial improvements over baselines.**

We performed an extensive suite of experiments on 59 correlated distributions, including 40 randomly generated, 7 hand-crafted and 12 adversarially generated correlated distributions. The revenue gap between the achieved revenue using our approach and the (unattainable) greedy upper bound is *maximally* $1.3\%$ over all non-adversarial distributions, and the gap goes up to $2.1\%$ under an evolutionary-computation generated adversarial distribution (A.6.1). Our performance is even more impressive in terms of the average revenue gap from the upper bound. For instance, in Table 1, the best baseline – the fully strategy-proof variant of RegretNet (Dütting et al., 2019), which is also based on our techniques – has an average revenue gap of $2.7\%$, while our neural network interpolation method achieves an average revenue gap of only $0.26\%$, representing a tenfold improvement!

Optimal auction for correlated bidders represents one of the most fundamental models in mechanism design. Based on our experiments, we can reasonably claim to have empirically solved this model.

**Contribution 4: Two examples demonstrating the generalisation capabilities of our techniques.**

In A.10, we extend our techniques to *multi-unit auctions with unit demand*, showing similarly strong performance. Myerson's greedy allocation is extended by assigning one item to each bidder with the highest non-negative virtual valuations. The extended monotonicity condition requires that any winner $i$, who is in the original *winner set*, must remains in the winner set when $i$ increases her bid while the other bids stay the same. We train a neural network to interpolate the extended greedy allocation and then verify whether the extended monotonicity condition holds.

In A.8, we show that it is convenient to integrate our techniques into RegretNet (Dütting et al., 2019) to design fully strategy-proof auctions, while the original RegretNet is only approximately strategy-

---

[5]Our monotonicity fix post-processes the allocation network to ensure strategy-proofness. GemNet (Wang et al., 2024) proposed a similar approach that post-processes the payment network to ensure menu compatibility (*i.e.*, preventing two bidders from winning the same item). Their method discretizes the bids and then extend to general bids via Lipschitz smoothness. In contrast, our technique operates directly on continuous values.

proof. RegretNet involves both the allocation and the payment networks. We require the allocation network be a tiny MLP with ReLU activation, so that we can apply our MIP-based techniques, including verification and post-processing monotonicity fix, to ensure that the final allocation is monotone. There is no restriction on the size or architecture of the payment network, as the payment network only serves as a surrogate, which will be thrown away when training ends. As long as the allocation is monotone, the "correct" payments can be reverse engineered from the allocation.

## 2 MODEL DESCRIPTION

We aim to design revenue-maximizing single-item auctions that are *deterministic*, *strategy-proof* and *ex post individually rational*. There are $n$ bidders. Bidder $i$'s valuation for the single item is denoted as $b_i$. Without loss of generality, we assume $0 \leq b_i \leq 1$. Since we focus on strategy-proof auctions, we do not differentiate between reported bids and private valuations. We use $\vec{b} = (b_1, b_2, \ldots, b_n)$ to denote the bid profile, which is drawn from a correlated distribution with the joint probability density function $\phi(\vec{b})$. We assume $\phi$ is continuous, everywhere positive, and bounded above by a constant. Our goal is to design an auction that maximizes expected revenue under $\phi$.

Following both Myerson (1981) and Papadimitriou & Pierrakos (2011)'s characterization, all deterministic, strategy-proof, and ex post individually rational mechanisms for our setting can be interpreted as *price-oriented rationing-free* mechanisms (Yokoo, 2003). That is, every bidder faces a deterministic take-it-or-leave-it offer, which potentially depends on the other bids. The offers must be structured so that for each bid profile, at most one bidder is willing to accept her offer.[6]

An allocation function $a$ maps each bid profile $\vec{b}$ to a binary vector $(a_1, a_2, \ldots, a_{n+1})$, where $a_i$ being 1 means bidder $i$ wins. The last dimension $a_{n+1}$ represents an auxiliary bidder, whose win results in the item not allocated. We require $\sum_i a_i = 1$. We use $a(\vec{b})_i$ to denote the $i$-th dimension of the allocation vector. Allocation monotonicity requires that $\forall i, b_{-i}, b_i, b_i'$ with $b_i < b_i'$, $a(b_i, b_{-i})_i \leq a(b_i', b_{-i})_i$. Myerson's characterization says that the allocation rule uniquely determines the payment rule. Using the take-it-or-leave-it offer interpretation, if bidder $i$ wins under profile $(b_i, b_{-i})$, then her payment must be exactly $\inf\{b_i' | a(b_i', b_{-i})_i = 1\}$. Losing bidders pay nothing.

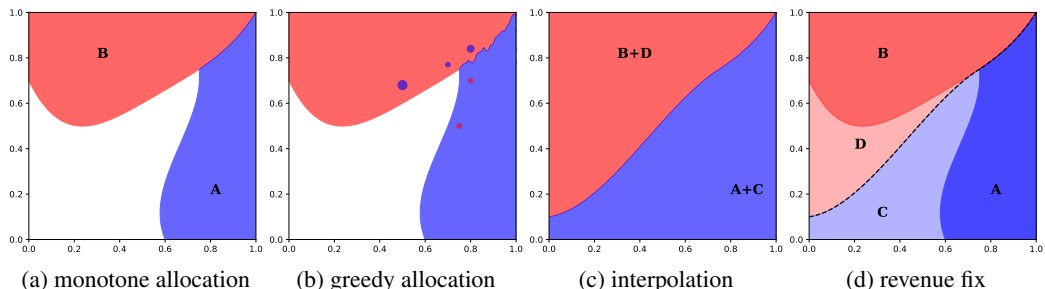

(a) monotone allocation    (b) greedy allocation    (c) interpolation    (d) revenue fix

Figure 1: (a) example monotone allocation; (b) example greedy allocation that is not monotone; (c) example monotone interpolation of (b), where the details are "glossed over" due to limited expressive capability of tiny networks; (d) final allocation after revenue fix on (c) (Section 3)

Figure 1a illustrates an example monotone allocation for 2 bidders. Every bid profile $(b_1, b_2) = (x, y)$ corresponds to a point in the unit square. Region A, i.e., the blue points, is where bidder 1 wins. Region B, i.e., the red points, is where bidder 2 wins. The white region is where the item is not allocated. Allocation monotonicity can be interpreted as follows: the blue region must be *rightward-closed*: for every blue point, all points on its right must also be blue. Similarly, the red region must be *upward-closed*: for every red point, all points above it must be red. The region boundaries characterize the payments. If $(x, y)$ is white, then the item is not allocated and no bidders pay. If $(x, y)$ is blue, then bidder 1 wins and her payment is the minimum $x'$ value so that $(x', y)$ is

---

[6]Considering that our objective is revenue in expectation and we assume a continuous probability density function bounded above by a constant, we can ignore all tie-breaking issues in this paper.

on the blue boundary. If $(x, y)$ is red, then bidder 2 wins and his payment is the minimum $y'$ value so that $(x, y')$ is on the red boundary.

# 3 Virtual Valuation, Ironed Virtual Valuation, and Marginal Profit

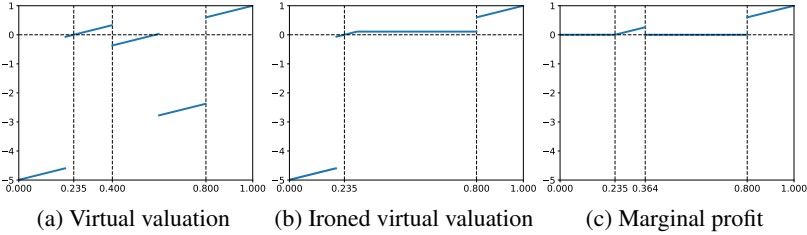

(a) Virtual valuation     (b) Ironed virtual valuation     (c) Marginal profit

Figure 2: Plots for grid distribution $[0.2, 3.58, 0.32, 0.05, 0.85]$. The grid distribution family will be formally defined in Section 5. The example grid distribution here describes a random variable with range $[0, 1]$, which is divided into 5 equally-sized sub-intervals. In $[0, \frac{1}{5}]$, the probability density value is uniformly $0.2$. In $[\frac{1}{5}, \frac{2}{5}]$, the probability density value is uniformly $3.58$ – etc.

Myerson (1981) introduced the *virtual valuation* and also a mathematical maneuver called *ironing*, leading to the *ironed virtual valuation*. Papadimitriou & Pierrakos (2011) introduced another variant called *marginal profit*, which comes with a technical change that is helpful for our neural interpolation approach. While we can define the greedy allocation based on any of these three variants, the marginal profit version consistently comes out on top in experiments described in Section 5. Due to space constraint, in this section, we only provide the bare minimum coverage of these concepts. More relevant details are deferred to A.2. Exposition of the virtual valuations, including alternative interpretations, can be found in Bulow & Roberts (1989); Fu (2017); Hartline (2013).

**Virtual valuation:** The virtual valuation is defined for independent bidders. Bidder $i$'s virtual valuation only depends on her own distribution and her own bid. We use $f_i$ and $F_i$ to denote the probability density function and the cumulative distribution function of bidder $i$'s valuation. As mentioned in Section 2, in our mechanism design context, all auctions can be interpreted as *price-oriented rationing-free* mechanisms (Yokoo, 2003), where every bidder faces a deterministic take-it-or-leave-it offer, which depends on the other bids. We use $o$ to denote the take-it-or-leave-it offer bidder $i$ faces, with the understanding that $o$ depends on the other bids $b_{-i}$ (but this turns out to be irrelevant in the mathematical analysis). Since it is a take-it-or-leave-it auction from the perspective of bidder $i$, the expected revenue we can extract from $i$ is $o(1 - F_i(o))$. The derivative of $o(1 - F_i(o))$ with respect to $o$ equals $1 - F_i(o) - of_i(o) = -(o - \frac{1 - F_i(o)}{f_i(o)})f_i(o)$. Since $F_i(1) = 1$, we can show

$$o(1 - F_i(o)) = \int_o^1 \left( x - \frac{1 - F_i(x)}{f_i(x)} \right) f_i(x) dx \tag{1}$$

The left side of the formula is the expected revenue we can extract from $i$. The right side can be interpreted as follows: Consider bidder $i$ who bids $x$. When $i$ wins, which is when $x \geq o$, her contribution to the revenue is $v_i(x) = x - \frac{1 - F_i(x)}{f_i(x)}$, which is called $i$'s virtual valuation.[7] When $i$ loses, which is when $x \leq o$, her contribution to the revenue is $0$ as $x$ is below the integration lower limit. Bidder $i$'s expected contribution in terms of virtual valuation equals the expected revenue from $i$. This interpretation leads to Myerson's greedy allocation. That is, we simply convert the bidders' original bids to their virtual valuations according to the above function $v_i$, and allocate the item to whoever has the highest virtual valuation. If the highest virtual valuation is negative, then we do not allocate. Being greedy, this allocation goes for the maximum revenue. The caveat is that it may not be monotone. Myerson calls a distribution *regular* if under it, $v_i$ is nondecreasing. If all bidders' distributions are regular and independent, then it is easy to show that the greedy allocation is monotone. If the winner increases her bid while the other bids stay the same, then due to distribution regularity, the winner's virtual valuation never decreases. The other bidders' virtual valuations do

---

[7]Figure 2a shows a sample plot, where the bid $x$ (x-axis) is mapped to the virtual valuation $v_i(x)$ (y-axis).

not change due to the independence assumption, so the winner still wins when she raises her bid while the other bids stay the same.

The virtual valuation can be extended to correlated distributions by switching to conditional probability functions. We use $v_i(b_i|b_{-i})$ to represent $i$'s virtual valuation when $i$'s bid is $b_i$ and the other bids are $b_{-i}$. The correlated version of virtual valuation is then $v_i(b_i|b_{-i}) = b_i - \frac{1-F_i(b_i|b_{-i})}{f_i(b_i|b_{-i})}$.

**Ironed virtual valuation:** The greedy allocation based on the virtual valuation guarantees monotonicity when the distribution is both regular and independent. Actually, the regularity assumption can be relaxed as long as independence holds. Myerson proposed an ironing technique that converts a not necessarily nondecreasing virtual valuation function (i.e., Figure 2a) to a nondecreasing version (i.e., Figure 2b). The effect of ironing is that the decreasing regions (and some of their nearby regions) become "flat", and within a flat region, the original virtual valuation is replaced by the average (weighted according to the probability density). Unfortunately, this averaging process is only mathematically sound with the independence assumption. While we can still iron in the presence of correlation, and base the greedy allocation on the ironed virtual valuations, the resulting auctions are suboptimal, as confirmed by Table 1. This is expected, as the averaging process is not meant to be optimal when there is correlation and replacing actual virtual valuations by averages causes "information loss". Considering that ironing does not align well with the correlated setting, we defer the details of the ironing process and its limitation in the presence of correlation to A.2.

**Marginal profit:** The final variant of the virtual valuation is the marginal profit (Papadimitriou & Pierrakos, 2011). Our presentation of the marginal profit is slightly modified from the original version, both for simplifying the presentation and for easier comparison against the virtual valuation. We present the original definition in A.2. We first present the marginal profit for independent bidders. Similar to virtual valuations, the correlated version can be obtained by switching to conditional distributions. We still use $f_i$ and $F_i$ to denote the probability density function and the cumulative distribution function of bidder $i$. The key difference between the marginal profit and the virtual valuation lies in the following *revenue fix* process.

*Revenue fix:* Given a monotone allocation, for any bidder $i$ and any set of other bids $b_{-i}$, $i$ faces a take-it-or-leave-it offer $o$, which depends on $b_{-i}$. We can raise the value of $o$ arbitrarily and this would never break monotonicity or cause over-allocation. Therefore, suppose we have an allocation that offers $o$ to bidder $i$ when the other bids are $b_{-i}$, instead of offering $o$ directly and achieve a revenue of $o(1 - F_i(o))$, we should *optimally* raise the offer to $\arg\max_{x \geq o} x(1 - F_i(x))$,[8] leading to a revenue of $\max_{x \geq o} x(1 - F_i(x))$.

The following equation defines the marginal profit, which is similar to Equation 1 for virtual valuation. In Equation 1, the left-hand side is the expected revenue extracted from bidder $i$. In Equation 2 below, the left-hand side is the expected revenue extracted from bidder $i$, after the revenue fix.

$$\max_{x \geq o} x(1 - F_i(x)) = \int_o^1 \left( -\frac{\partial(\max_{x' \geq x} x'(1 - F_i(x')))}{\partial x} / f_i(x) \right) f_i(x) dx \qquad (2)$$

We define $m_i(x) = -\frac{\partial(\max_{x' \geq x} x'(1 - F_i(x')))}{\partial x} / f_i(x)$ to be the marginal profit of bidder $i$ when her bid is $x$. The message of Equation 1 is that the expected revenue is equal to the expected virtual valuation. Equation 2 conveys a similar message, which is that the expected revenue *after the revenue fix* is equal to the expected marginal profit.

**Virtual valuation versus marginal profit:** If the distribution is regular, then the derivative of $x(1 - F_i(x))$ with respect to $x$, i.e., the original virtual valuation, is nondecreasing, which means that there is a cutoff value $x^0$, so that $x(1 - F_i(x))$ is nonincreasing when $x \leq x^0$ and it is nondecreasing when $x \geq x^0$. This implies $m_i(x) = 0$ if $x \leq x^0$ and $m_i(x) = v_i(x)$ if $x \geq x^0$. That is, for regular distributions, $m_i(x) = \max\{v_i(x), 0\}$. The only difference is the additional "max". For general distributions, the difference is more notable, but it remains based on the same underlying principle.

Although the two concepts are minor variations of each other, interpolating based on marginal profit offers certain advantages, as confirmed by experiments in Table 1. Below, we provide an example to illustrate the benefits of marginal profit over virtual valuation. We take the uniform distribution over $[0, 1]$ as an example, which is regular. Suppose we only have one bidder. The optimal offer should

---

[8]$\arg\max_{x \geq o} x(1 - F_i(x))$ may be exactly $o$, in which case we keep the original offer.

be 0.5, which can be interpreted as an optimal reserve price. Myerson's virtual valuation function is $v_i(x) = x - \frac{1 - F_i(x)}{f_i(x)} = 2x - 1$. The marginal profit function is $m_i(x) = \max\{2x - 1, 0\}$. According to the virtual valuation, when the bid $x$ is below 0.5, the virtual valuation is negative, which means we should allocate the item only if the bid is at least 0.5, therefore explaining the optimal reserve at 0.5. On the other hand, the marginal profit is 0 when the bid $x$ is below 0.5. Therefore, when the bid is lower than 0.5, allocating is neither helpful nor detrimental by this measure. When the bid is higher than 0.5, the marginal profit is positive, so allocating is helpful. So according to marginal profit, every offer in $[0, 0.5]$ is optimal, including offering 0. *Actually* offering $o < 0.5$ is certainly not optimal, but such a suboptimal offer can be easily fixed by raising it to the best revenue point, i.e., to $\arg\max_{x \geq o} x(1 - F_i(x)) = 0.5$. Another way to put it is that the suboptimal offer can be fixed by pushing it up until the marginal profit becomes positive, i.e., to $\inf\{x | m_i(x) > 0, x \geq o\} = 0.5$. The above example illustrates an advantage of marginal profit – it cuts more slack to our learning procedure. If we learn the allocation for this example by mimicking the greedy allocation based on either the virtual valuation or the marginal profit, then if we go with the virtual valuation, the learned reserve must be *exactly* 0.5. Any more or less is considered not optimal. On the other hand, according to the marginal profit, 0.43 is optimal, and so is 0.37. The exact optimal reserve can always be recovered by pushing up the offer via the revenue fix process. It should be noted that we *must* go through the trouble of performing the revenue fix process if the learning target is based on the marginal profit. Otherwise, we run the risk of using a reserve of 0, which is far from optimal.

We conclude this section with an example illustrating the revenue fix process.

**Revenue fix example:** We refer to Figure 1b, which shows an example greedy allocation that is not monotone. Blue points are when bidder 1's marginal profits are strictly higher. Red points are when bidder 2's marginal profits are strictly higher. White points are when both bidders' marginal profits are zero. We train a neural network to fit the greedy allocation, resulting in a monotone allocation depicted in Figure 1c. Since our network size is tiny, it glosses over the fine details, thereby avoiding the minor monotonicity violations by the greedy allocation (i.e., the blue points enclosed by the red region and the red points enclosed by the blue region). We note that when a point's marginal profit is 0, it does not matter which bidder wins. In Figure 1c, the network simply allocates the white points arbitrarily. That is, the focus of interpolation is on regions that actually matter (i.e., the limited expressive power of our tiny network focuses on creating a separation between A and B). Figure 1d shows the aftermath of the revenue fix. The learned allocation allocates A+C to bidder 1 and B+D to bidder 2. After the fix, C and D become unallocated. The final allocation is the one in Figure 1a.

## 4 Technical Description of the Proposed Approach

We train a neural network to mimic the greedy allocation, which has three versions, depending on whether we greedily allocate according to the vanilla virtual valuation, the ironed virtual valuation, or the marginal profit. When there is no ambiguity, greedy allocation refers to the version based on marginal profit. Even though we are maximizing revenue, we do not need to reference any payment function in our training. Training is solely carried out on the allocation function.

We represent the allocation rule as an MLP with ReLU activation. The inputs are the bids ($n$ dimensions). The output dimension is $n + 1$. If the $i$-th ($i = 1, 2, \ldots, n$) output coordinate is the highest, then bidder $i$ wins. If the $(n + 1)$-th coordinate is the highest, then the item is not allocated.

In Myerson's original approach, during the derivation of the optimal auction, randomization is allowed, which simplifies the mathematical analysis. Similarly, we allow randomization to facilitate the training. We apply softmax to the network's outputs, ensuring that the coordinate with the highest value corresponds to the highest proportion of the item won in the context of randomized auctions. During training, we sample a batch of bid profiles. For each profile $\vec{b}$, we calculate the marginal profits of each bidder. The marginal profit of the auxiliary bidder $(n + 1)$ is always 0. We use $m(\vec{b})$ to represent the vector of marginal profits. We use *NN* to represent the network. The training loss is simply the batched sum of $-m(\vec{b}) \cdot \text{softmax}(NN(\vec{b}))$ over all training samples in the batch. This mimics greedy allocation because in the case of perfect fit, if the marginal profit of bidder $i$ is the highest, then the $i$-th coordinate of $\text{softmax}(NN(\vec{b}))$ should be 1 and the other coordinates should be 0's. During evaluation, we revert back to the deterministic version. That is, we do not apply softmax and simply pick the highest output coordinate as the winner.

Our techniques also include the use of two mixed-integer programs. The first is for verifying whether the trained allocation is monotone. For a trained network, its weights and biases are viewed as constants. Since our network is based on MLP+ReLU, for any node in the network, its value before activation can be written as a linear expression involving the activated versions of the node values from the previous layer. The ReLU activation step can be modelled using the big-M trick with the introduction of an auxiliary binary variable. When presenting the mixed-integer programs, we simply use $(a_1, a_2, \ldots, a_{n+1}) = NN(\vec{b})$ to represent that the $a_i$'s are the outputs of the network when the input is $\vec{b}$, with the understanding that the $a_i$ can be written as linear expressions of $\vec{b}$ with the help of auxiliary binary variables. The MIP for monotonicity verification finds the largest gap between $b_i$ and $b_i'$ so that $i$ wins when bidding $b_i$, but after increasing her bid to $b_i'$ while keeping the other bids fixed, a different bidder $j$ wins ($j$ can be $n + 1$). The gap is the maximum over $\forall b_{-i}$. In order to verify monotonicity, we need to run $n^2$ MIPs (for every $i$ from 1 to $n$ and for every $j \neq i$ from 1 to $n + 1$). All MIPs' objectives must be 0's (or infeasible) to conclude monotonicity.

**MIP for monotonicity verification**

| Constants | $1 \leq i \leq n; j \neq i; 1 \leq j \leq n + 1$ |
|---|---|
| Variables | $0 \leq b_1, b_2, \ldots, b_n, b_i' \leq 1; b_i \leq b_i'$ |
| Maximize | $b_i' - b_i$ |
| subject to | $(a_1, a_2, \ldots, a_{n+1}) = NN(b_i, b_{-i})$ |
| | $a_i = \max_t a_t$ |
| | $(a_1', a_2', \ldots, a_{n+1}') = NN(b_i', b_{-i})$ |
| | $a_j' = \max_t a_t'$ |

**MIP for monotonicity fix**

| Constants | $1 \leq i \leq n; j \neq i; 1 \leq j \leq n + 1$ |
|---|---|
| | $0 \leq b_1, \ldots, b_{i-1}, b_{i+1}, \ldots, b_n \leq 1$ |
| Variables | $0 \leq b_i' \leq 1$ |
| Maximize | $b_i'$ |
| subject to | $(a_1, a_2, \ldots, a_{n+1}) = NN(b_i', b_{-i})$ |
| | $a_j = \max_t a_t$ |

The by-products of the MIP for monotonicity verification are the counterexamples. Given a counterexample, we add a penalty term $\lambda \cdot \text{ReLU}(\text{softmax}\,(NN(b_i, b_{-i}))_i - \text{softmax}(NN(b_i', b_{-i}))_i)$, which aims to ensure that by increasing her bid, bidder $i$'s proportion of item won must never decrease.

With additional training on counterexamples, we sometimes, but not always, end up with verifiably monotone allocations. When we do not manage to achieve verifiable monotonicity, we can always apply a post-processing monotonicity fix using the second MIP. This fix is applied after collecting the bids. When calculating $i$'s fix, the bids in $b_{-i}$ are treated as constants. For bidder $i$, the second MIP finds the maximum bid for $i$ where $i$ loses (according to the trained network) to a different bidder $j$ ($j$ could be $n + 1$). We need to solve $n$ MIPs only for the (tentatively, i.e., according to the trained network) winning bidder, and the maximum over the $n$ objective values (ignoring infeasible MIPs) is the monotonicity cutoff price for $i$. As $i$'s bid goes from this cutoff price to 1, $i$ always wins, which implies monotonicity. But if $i$'s bid was below this, $i$ does not win after all. The monotonicity fix can be applied together with the revenue fix from Section 3. For example, let the original offer produced by the network be $o$. The revenue fix instructs to raise the offer to $o_r$ and the monotonicity fix instructs to raise the offer to $o_m$. The final offer is then $\max\{o, o_r, o_m\}$.

## 5   EXPERIMENTS

$$\begin{bmatrix} 11 & 24 & 7 & 20 & 3 \\ 4 & 12 & 25 & 8 & 16 \\ 17 & 5 & 13 & 21 & 9 \\ 10 & 18 & 1 & 14 & 22 \\ 23 & 6 & 19 & 2 & 15 \end{bmatrix}$$

To facilitate experiments, we introduce the *grid distributions*, where we use $n$-dimensional matrices to represent correlated distributions for $n$ bidders. The example 2D matrix on the right represents a correlated distribution for 2 bidders. Here, the matrix size is $5 \times 5$. We divide the unit square $[0, 1] \times [0, 1]$ into 25 sub-squares of side length $\frac{1}{5}$. The bottom left element is 23, which means that when the bid profile $(x, y)$ falls into sub-square $[0, \frac{1}{5}] \times [0, \frac{1}{5}]$, the joint probability density is *uniformly* 23. The element on the right of 23 is 6, which means that for the sub-square $[\frac{1}{5}, \frac{2}{5}] \times [0, \frac{1}{5}]$, the joint probability density is uniformly 6.[9]

---

[9] There is an additional normalization ratio, which will be multiplied with every density value to ensure that the total probability equals exactly 1. To make the presentation cleaner, we omit this normalization ratio.

There are multiple reasons for introducing the grid distributions. 1) They allow the *systematic generation* of correlated distributions. 2) All relevant numerical operations can be implemented *analytically*, including derivative, maximum, and integration. This is important for producing *numerically stable* experimental results as the marginal profit is the *partial derivative* of the *maximum* of a term involving the cumulative distribution function, which calls for an *integration*. 3) Theoretically speaking, the grid distributions can approximate *any* correlated distribution if we allow arbitrarily fine grids. 4) The grid distribution family makes it easy to generate *adversarial distributions* via tuning the matrix. For example, in A.6, we report on experiments using evolutionary computation to evolve an adversarial matrix, as well as generating adversarial matrices using mixed-integer-programming. 5) We can easily construct hand-crafted grid distributions. For example, we use *magic squares* – 2D matrices in which every row and every column sums to the same value – to create correlated distributions where each bidder's marginal distribution is uniform. The matrix shown above is a magic square. We use magic-square-based distributions to evaluate the effect of correlation on revenue. For such distributions, if we ignore correlation, then the distribution becomes i.i.d. uniform, where the second price auction with reserve is optimal. As a result, on correlated distributions, the gap between our revenue and the revenue from a second-price auction with a reserve price based on an i.i.d. distribution represents the additional revenue we can gain by considering correlation in auction design.

We altogether experimented with 59 different grid distributions, including

- 40 randomly generated grid distributions: For $n$ from 2 to 5, we generate 10 random grid distributions of size $\underbrace{5 \times \ldots \times 5}_{n}$, where every matrix element is drawn from uniform 0 to 1. We use $G5_s$ to denote the distribution generated with seed $s$.

- 7 hand-crafted grid distributions: These are based on magic squares described in A.5.

- 12 adversarially generated distributions: 10 were generated using evolutionary computation and 2 were generated using mixed-integer-programming. We defer all discussion on adversarial distributions to A.6. *We use them to test the limit of our proposed approach.*

As mentioned in Section 3, there are three variants of the virtual valuation, leading to three greedy allocations that can be used as interpolation targets. For 2 bidders, we compare the revenue achieved using our approach based on MP (marginal profit), VV (vanilla virtual valuation) and IVV (ironed virtual valuation). MP consistently comes out on top. For 3 to 5 bidders, we only focus on MP. Our experiments also involve the following baseline auctions (detailed descriptions are in A.4):

- GREEDY (based on marginal profit): For 56 out of the 59 distributions, the greedy allocation is proven not monotone via counterexamples. Thus, GREEDY violates the key constraint and only serves as the revenue *upper bound*.

- AMD: We round the bids down to the nearest multiple of 0.01 and apply the classical approach to automated mechanism design (Conitzer & Sandholm, 2002; 2004), which is a mixed-integer-programming-based approach producing the optimal auction when we restrict to discrete bids. We use AMD as an alternative implementation to the FPTAS for 2 correlated bidders from Papadimitriou & Pierrakos (2011), which is also based on discrete bids. Both AMD and FPTAS are for 2 bidders only: AMD does not scale beyond 2 bidders for our setting and the FPTAS only applies to 2 bidders (the underlying model becomes NP-hard to approximate for 3 bidders).

- MYERSON: Myerson's optimal auction when ignoring correlation. The bidders' ironed virtual valuations are based on the *marginal* distributions (instead of the *conditional* distributions).

- 2ND: Second price auction with the optimal reserve.

- RNET: Integration of verification to RegretNet (Dütting et al., 2019), details in A.8. We model the allocation network via a tiny MLP with ReLU activation, so that we can apply our MIP-based verification to ensure that the final allocation is monotone, which makes our resulting auctions fully strategy-proof. (The original RegretNet approximates strategy-proofness.) The payment network, without any architectural restrictions, only serves as a surrogate, which is thrown away when training ends. The "correct" payments are reverse engineered from the verifiably monotone allocation.

In the following tables, we present the summary results for $n \in \{2, 3, 5\}$. The full result tables including standard errors, results for $n = 4$, and results for adversarial distributions are deferred to Appendix A.11. A value is bold if it is the best among the scalable approaches (AMD is not in the comparison). MP consistently achieves the best revenue among the scalable approaches. When MP also outperforms AMD, we mark the result with a '*'. Sometimes MP's value appears to be higher than the upper bound GREEDY. This is because the numbers shown are based on Monte Carlo simulation (see the full tables with standard errors in A.11 and the evaluation details in A.3).

GREEDY's revenue is an unattainable upper bound. We use GAP(METHOD) to denote the average revenue gap between METHOD and GREEDY, formally defined as

$$\text{GAP(METHOD)} = \frac{\text{Average revenue of GREEDY} - \text{Average revenue of METHOD}}{\text{Average revenue of GREEDY}}$$

In Table 1, the best baseline RNET – the fully strategy-proof variant of RegretNet (Dütting et al., 2019), which is also based on our techniques – has an average revenue gap of $2.7\%$, while our method MP achieves an average revenue gap of only $0.26\%$, representing a tenfold improvement!

Table 1: 2 bidders
$\text{GAP(MP)} = 0.26\%, \text{GAP(RNET)} = 2.7\%, \text{GAP(2ND)} = 4.2\%, \text{GAP(MYER)} = 4.3\%$

| DISTRIB. | GREEDY | AMD | MYER. | 2ND | RNET | MP | VV | IVV |
|---|---|---|---|---|---|---|---|---|
| $G5_0$ | 0.4353 | 0.4368 | 0.4191 | 0.4197 | 0.4307 | **0.4368** | 0.4355 | 0.4329 |
| $G5_1$ | 0.4047 | 0.4041 | 0.3939 | 0.3929 | 0.4021 | **0.4037** | 0.4032 | 0.4032 |
| $G5_2$ | 0.3979 | 0.3967 | 0.3771 | 0.3823 | 0.3869 | **0.3959** | 0.3946 | 0.3955 |
| $G5_3$ | 0.4629 | 0.4625 | 0.4533 | 0.4546 | 0.4522 | **0.4628*** | 0.4612 | 0.4616 |
| $G5_4$ | 0.4681 | 0.4665 | 0.4476 | 0.4472 | 0.4639 | **0.4674*** | 0.4673 | 0.4671 |
| $G5_5$ | 0.4204 | 0.4187 | 0.4075 | 0.4057 | 0.4062 | **0.4151** | 0.4148 | 0.4150 |
| $G5_6$ | 0.3905 | 0.3912 | 0.3761 | 0.3741 | 0.3798 | **0.3908** | 0.3893 | 0.3886 |
| $G5_7$ | 0.4865 | 0.4854 | 0.4494 | 0.4544 | 0.4771 | **0.4850** | 0.4828 | 0.4840 |
| $G5_8$ | 0.4298 | 0.4273 | 0.3988 | 0.3981 | 0.4077 | **0.4291*** | 0.4281 | 0.4279 |
| $G5_9$ | 0.4531 | 0.4512 | 0.4306 | 0.4302 | 0.4379 | **0.4498** | 0.4482 | 0.4497 |
| SATURN | 0.4533 | 0.4545 | 0.4210 | 0.4226 | 0.4372 | 0.4510 | 0.4512 | **0.4515** |
| JUPITER | 0.4427 | 0.4404 | 0.4238 | 0.4219 | 0.4316 | **0.4418*** | 0.4409 | 0.4410 |
| MARS | 0.4375 | 0.4377 | 0.4177 | 0.4167 | 0.4194 | **0.4360** | 0.4333 | 0.4359 |
| SOL | 0.4429 | 0.4414 | 0.4376 | 0.4364 | 0.4329 | **0.4427*** | 0.4404 | 0.4419 |
| VENUS | 0.4381 | 0.4381 | 0.4175 | 0.4171 | 0.4233 | **0.4365** | 0.4319 | 0.4358 |
| MERCURY | 0.4273 | 0.4274 | 0.4179 | 0.4176 | 0.4150 | **0.4281*** | 0.4271 | 0.4279 |
| LUNA | 0.4331 | 0.4335 | 0.4173 | 0.4172 | 0.4212 | **0.4322** | 0.4315 | 0.4318 |

Table 2: 3 and 5 bidders: M/2 represents the better between MYERSON and 2ND
$n = 3$: $\text{GAP(MP)} = 0.36\%, \text{GAP(RNET)} = 3.6\%, \text{GAP(2ND)} = 3.9\%, \text{GAP(MYER)} = 3.8\%$
$n = 5$: $\text{GAP(MP)} = 0.87\%, \text{GAP(RNET)} = 2.8\%, \text{GAP(2ND)} = 2.4\%, \text{GAP(MYER)} = 2.4\%$

| | $n = 3$ | | | | $n = 5$ | | | |
|---|---|---|---|---|---|---|---|---|
| DISTRIB. | GREEDY | M/2 | RNET | MP | GREEDY | M/2 | RNET | MP |
| $G5_0$ | 0.5512 | 0.5346 | 0.5346 | **0.5517** | 0.6862 | 0.6700 | 0.6670 | **0.6799** |
| $G5_1$ | 0.5647 | 0.5475 | 0.5509 | **0.5651** | 0.6901 | 0.6738 | 0.6713 | **0.6846** |
| $G5_2$ | 0.5316 | 0.5137 | 0.5153 | **0.5302** | 0.6838 | 0.6709 | 0.6681 | **0.6783** |
| $G5_3$ | 0.5476 | 0.5283 | 0.5290 | **0.5450** | 0.6894 | 0.6724 | 0.6684 | **0.6838** |
| $G5_4$ | 0.5485 | 0.5306 | 0.5310 | **0.5471** | 0.6853 | 0.6692 | 0.6664 | **0.6787** |
| $G5_5$ | 0.5477 | 0.5237 | 0.5255 | **0.5443** | 0.6850 | 0.6720 | 0.6685 | **0.6809** |
| $G5_6$ | 0.5371 | 0.5155 | 0.5158 | **0.5332** | 0.6902 | 0.6725 | 0.6686 | **0.6839** |
| $G5_7$ | 0.5723 | 0.5528 | 0.5555 | **0.5710** | 0.6900 | 0.6719 | 0.6682 | **0.6824** |
| $G5_8$ | 0.5473 | 0.5193 | 0.5185 | **0.5429** | 0.6868 | 0.6715 | 0.6659 | **0.6807** |
| $G5_9$ | 0.5663 | 0.5410 | 0.5396 | **0.5637** | 0.6927 | 0.6768 | 0.6711 | **0.6866** |

In A.10, we extend to multi-unit auctions with unit demand, where we achieved similar strong performance. In Table 6, the baseline $(m + 1)$-th price auction with the optimal reserve has an average revenue gap of $5.1\%$, while our method MP has an average revenue gap of $0.67\%$.

REPRODUCIBILITY STATEMENT

Training parameters, evaluation details, and hardware specifications are detailed in A.3. The code is included as part of the submission.

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

# A Appendix

## A.1 Related Research

Papadimitriou & Pierrakos (2011) proposed the marginal profit, a variant of Myerson's virtual valuation. The detailed comparison between virtual valuation and marginal profit is in Section 3. Papadimitriou & Pierrakos (2011) showed that for two correlated bidders, there exists a FPTAS for deriving the optimal auction, but for three correlated bidders, the problem becomes NP-hard to approximate. The FPTAS for two bidders first rounds the bids down to discrete grid points, which is then solved as a maximum independent set instance. Dobzinski et al. (2011) showed that if we switch to truthful-in-expectation, then the optimal auction for three or more correlated bidders can be found in polynomial time. However, the authors' definition of "polynomial time" is with respect to the size of the support of the correlated distribution, which is infinite for continuous bids and exponential (in the number of bidders) for discrete bids. Myerson (1981)'s original work showed that for independent bidders, there exists an optimal deterministic auction even when randomized auctions are allowed. On the other hand, Caragiannis et al. (2016) showed that, for correlated bidders, requiring determinism does incur a revenue loss. Roughgarden & Talgam-Cohen (2013) proposed a theoretical condition that is sufficient for showing that Myerson's greedy allocation is still monotone for correlated bidders. On the other hand, it is not clear which distributions satisfy the proposed theoretical condition and given a specific distribution, it is not clear how to computationally validate whether the theoretical condition holds. For 56 out of 59 distributions we experimented on in this paper, Myerson's greedy allocation is not monotone, and for the remaining 3 distributions, monotonicity is inconclusive, i.e., all we can say is that counterexamples have not be found. Ronen (2001) proposed an approximate auction that guarantees half of the revenue for correlated bidders. Crémer & McLean (1988) showed that it is possible to extract the full social surplus for correlated bidders, but the proposed auction is only individually rational in expectation. Bei et al. (2019) studied single-item auctions for correlated bidders from the lens of worst-case analysis. The authors evaluate an auction based on its revenue under the worst-case correlation.

This paper follows the line of research on neural network mechanism design initiated by Dütting et al. (2019). Curry et al. (2020) first applied mixed-integer-programming to evaluate strategy-proofness of neural-network-based mechanisms. Guo (2024) proposed a suite of techniques on worst-case mechanism design via neural networks, including worst-case counterexample-guided training where the worst-case profiles are obtained via mixed-integer-programming and using the Lottery Ticket Hypothesis (Frankle & Carbin, 2019) to guide the search for tiny networks that are trainable for worst-case training objectives. Duan et al. (2024) studied the design of *virtual valuation combinatorial auctions* originally proposed in Sandholm & Likhodedov (2015) via neural network training. Despite the name, the "virtual valuation" in virtual valuation combinatorial auctions is a different concept, which is loosely inspired by Myerson's original concept of virtual valuation. The monotonicity fix in our paper post-processes the allocation network to ensure strategy-proofness. GemNet (Wang et al., 2024) proposed a similar approach that post-processes the payment network to ensure menu compatibility (*i.e.*, preventing two bidders from winning the same item). Their method discretizes the bids and then extend to general bids via Lipschitz smoothness. In contrast, our technique operates directly on continuous values.

While there are works on enforcing monotonicity via special neural network structures (Sill, 1997; Milani Fard et al., 2016), these monotonicity networks are regarding the monotonicity of the network outputs, which is different from allocation monotonicity. For all distributions we experimented on, we manage to achieve verifiable monotonicity using standard MLP with ReLU activation.

## A.2 IRONING AND MARGINAL PROFIT

Here, we present a high-level overview of Myerson's ironing approach, which requires the independence assumption. We still use $f_i$ and $F_i$ to denote bidder $i$'s probability density function and cumulative distribution function. Myerson's ironing approach requires the introduction of new notations. We use $q = 1 - F_i(o)$ to denote bidder $i$'s *demand* when faced with offer $o$. We use $R_i(q)$ to denote the revenue extracted from bidder $i$ when her demand is $q$. That is, $R_i(q) = qo = qF_i^{-1}(1-q)$. The derivative of this revenue with respect to the demand is then $\frac{dR_i(q)}{dq} = F_i^{-1}(1-q) - \frac{q}{f_i(F_i^{-1}(1-q))}$.

If we rewrite this derivative in terms of $o$, noting that $o = F_i^{-1}(1-q)$, then the derivative becomes $o - \frac{1-F_i(o)}{f_i(o)}$, which is exactly $v_i(o)$, i.e., the virtual valuation at value $o$. If the virtual valuation is nondecreasing in $o$, then it is nonincreasing in $q$, as higher offers correspond to lower demands. This implies that for regular distributions, $R_i(q)$ is *concave* (i.e., its derivative does not increase in $q$). For non-regular distributions, $R_i(q)$ is not concave, which means that there exist two demand values $q_1$ and $q_2$, so that $\frac{R_i(q_1)+R_i(q_2)}{2} \geq R_i(\frac{q_1+q_2}{2})$. During Myerson's derivation of the optimal auction, randomized auctions are allowed. It is just that the final optimal auction is proven to be deterministic. When randomized auctions are allowed, the offer leading to demand $\frac{q_1+q_2}{2}$ is never a good idea, because more revenue can be achieved by replacing this offer by a half/half mixture of the offers leading to demand $q_1$ and $q_2$. In summary, Myerson showed that we can always resort to randomization to achieve a concave revenue function $\overline{R_i}(q)$ on the basis of the original $R_i(q)$, i.e., by pushing up $R_i(\frac{q_1+q_2}{2})$ to $\frac{R_i(q_1)+R_i(q_2)}{2}$ for the non-concave regions. In terms of implementation, one option is the Graham scan algorithm, a computational geometry algorithm for generating the convex hull given a set of points. Recall that the derivative of the original revenue function $R_i(q)$ is the original virtual valuation. Myerson uses the derivative of $\overline{R_i}(q)$ to serve as the ironed virtual valuation, which is guaranteed to be nondecreasing (in the bid) due to the concavity of $\overline{R_i}(q)$. Note that $R_i$ and $\overline{R_i}$ would differ in one or several regions. For example, let $[q_1, q_2]$ be one region where $R_i$ and $\overline{R_i}$ differ. The curve of $\overline{R_i}$ on $[q_1, q_2]$ is exactly the straight line connecting $(q_1, R_i(q_1))$ and $(q_2, R_i(q_2))$, which means that the derivative of $\overline{R_i}$, i.e., the ironed virtual valuation, must be a constant in $[q_1, q_2]$. This is how the flat region in Figure 2b is calculated.

Next, we discuss why the above ironing process is no longer valid for correlated distributions and why it causes "information loss" in training. In Figure 2b, there is a long flat region $[0.289, 0.8]$. The virtual valuation in $[0.289, 0.8]$ is replaced by the average $0.109$. Suppose that the curve shown in Figure 2b is for bidder $i$. For independent bidders, when we apply the greedy allocation, if the other bidders' ironed virtual valuations do not exceed $0.109$, then bidder $i$ would win if her bid is at least $0.289$, which makes it correct to use the average to replace the actual virtual valuations, because bidder $i$ "wins" for the whole interval $[0.289, 0.8]$. If the maximum of the other bidders' ironed virtual valuations exceeds $0.109$, then bidder $i$ does not win with any bid below $0.8$, so the virtual valuation changes in $[0.289, 0.8]$ (due the averaging process) are irrelevant. When we extend to correlated distributions (by defining ironed virtual valuations in terms of conditional distributions), the above analysis no longer holds. When bidder $i$ changes her bid within $[0.289, 0.8]$, the other bidders' ironed virtual valuations may change due to correlation. For example, it is possible that when bidder $i$ bids below $0.4$, there is another bidder whose virtual valuation is higher than $0.109$, and when bidder $i$ bids above $0.4$, all other bidders' virtual valuations are below $0.109$. In this situation, within the flat region, bidder $i$ would only win in the region $[0.4, 0.8]$. The ironed virtual valuation suggests that getting allocated in $[0.4, 0.8]$ is beneficial to the revenue, as this flat region is positive in Figure 2b. But according to the actual virtual valuation in Figure 2a, getting allocated in $[0.4, 0.8]$ actually hurts the revenue.

Finally, we present the original definition of marginal profit from Papadimitriou & Pierrakos (2011) and compare it against the way we present it in Section 3. We use $m_i(b_i|b_{-i})$ to denote the marginal profit of bidder $i$ when her bid is $b_i$ and the other bids are $b_{-i}$. Recall that $\phi$ is the joint probability density function. We use $\varphi(b_{-i})$ to denote the joint probability density function of the bids except for $i$'s bid. The original definition in Papadimitriou & Pierrakos (2011) is

$$m_i(b_i|b_{-i}) = -\frac{\partial \left( \max_{x' \geq b_i} x' \int_{x'}^1 \phi(t, b_{-i}) dt \right)}{\partial b_i}.$$

We can rewrite it as follows:

$$
\begin{aligned}
m_i(b_i|b_{-i}) &= -\varphi(b_{-i}) \frac{\partial \left( \max_{x' \geq b_i} x' \int_{x'}^1 f_i(t|b_{-i})dt \right)}{\partial b_i} \\
&= -\varphi(b_{-i}) \frac{\partial \left( \max_{x' \geq b_i} x'(1 - F_i(x'|b_{-i})) \right)}{\partial b_i} \\
&= -f_i(b_i|b_{-i})\varphi(b_{-i}) \frac{\partial \left( \max_{x' \geq b_i} x'(1 - F_i(x'|b_{-i})) \right)}{\partial b_i} \Big/ f_i(b_i|b_{-i}) \\
&= -\phi(\vec{b}) \frac{\partial \left( \max_{x' \geq b_i} x'(1 - F_i(x'|b_{-i})) \right)}{\partial b_i} \Big/ f_i(b_i|b_{-i}).
\end{aligned}
$$

Our presentation (after switching to conditional distributions) is

$$
m_i(b_i|b_{-i}) = -\frac{\partial \left( \max_{x' \geq b_i} x'(1 - F_i(x'|b_{-i})) \right)}{\partial b_i} \Big/ f_i(b_i|b_{-i}).
$$

The only difference is the multiplier $\phi(\vec{b})$. That is, our presentation of the marginal profit is essentially the original marginal profit "per density". Since what we truly care about is the relative order of the marginal profits, i.e., which bidder has the highest marginal profit, the two presentations are equivalent for this purpose. We decide to go with the "per density" version as the original virtual valuation is "per density". Our presentation enabled the direct comparison between the marginal profit and the (ironed) virtual valuation in Figure 2.

### A.3 TRAINING PARAMETERS, EVALUATION DETAILS, AND HARDWARE

The description below is for our main approach. For integration with RegretNet, please refer to A.8.

**Training:** We use an MLP with 2 hidden layers to represent the allocation function. Each layer contains 20 nodes. We use the Adam optimizer with learning rate 0.001. Unless otherwise specified, we allow the optimizer to step 20,000 times. The batch size is 16. For counterexample-guided training, we use all counterexamples and the penalty term is multiplied by 1,000.

The following description applies to all distributions except for WORST10 and WORST100 (A.6.2). For 2 bidders, we only need to train once to achieve a monotone allocation. No counterexample-guided training or monotonicity fix is needed. This is true even for the 10 adversarial distributions generated using evolutionary computation (A.6.1). For 3 to 5 bidders, we train 10 times as some trials failed to achieve monotonicity. 10 trials are more than enough. Even in the worst situation, at least 6 of the 10 trials ended up monotone without using counterexample-guided training or monotonicity fix. For WORST10 and WORST100, we do need counterexample-guided training and monotonicity fix. Otherwise, near-optimal revenue cannot be achieved. The experiments on these two distributions are presented in A.6.2.

**Evaluation:** We evaluate an auction's revenue via Monte Carlo average with sample size 100,000. The full result tables in A.11 include the standard errors. When monotonicity fix is needed, the sample size is reduced to 10,000, as for each profile, monotonicity fix involves $n$ mixed-integer programs.

**Hardware:** The only large-scale experiments are the experiments described in Figure 3, Figure 4 and Figure 5, where for each of the three selected distributions ($G5_6$, WORST10, WORST100), we performed 100 training trails. This is carried out on University of Anonymity's high-performance cluster. Our performance bottleneck is the MIPs for monotonicity fix, which needs to be repeated $n \times 10,000$ times in evaluation, so our jobs are CPU intensive. (Running the auction once is always instant.) For each of the 300 trials, we allocate one CPU core, 128 GB memory, and no GPU. The CPU type is Intel(R) Xeon(R) Platinum 8360Y. Each trial takes from a few minutes (when monotonicity fix is not needed) to up to six hours. We have to point out that the wall clock time is highly inaccurate for the cluster we use (i.e., CPUs are shared; jobs may hang or slow down without notice). We estimate that the total computation time is on the order of 500 hours.

A.4 DETAILS OF BASELINE AUCTIONS

GREEDY: The greedy allocation simply allocates the item to the bidder with the highest marginal profit. Based directly on the definition of marginal profit (Papadimitriou & Pierrakos, 2011), the greedy allocation's revenue can serve as the revenue *upper bound*. To calculate this upper bound, we use Monte Carlo simulation to calculate the expectation of the highest marginal profit. For 56 of the 59 distributions we use, the greedy allocation is **not** monotone, and therefore **not** strategy-proof, which is proved via counterexamples. For the remaining 3 distributions, monotonicity/strategy-proofness is inconclusive.

AMD: Papadimitriou & Pierrakos (2011) proposed a FPTAS for deriving the optimal auction for 2 correlated bidders. The proposed algorithm first rounds down the bids to grid points, for example, to multiples of $0.01$. The rest of the algorithm converts the instance to a maximum independent set instance on a bipartite graph. We use AMD (Automated Mechanism Design (Conitzer & Sandholm, 2002)) as an alternative implementation. We round the bids down to multiples of $0.01$. After this step, there are $100^2$ possible bid profiles. For each bid profile, we create 2 continuous variables to represent the payments and 2 binary variables to represent the allocation. We follow the standard AMD process. The expected revenue can be easily expressed as a linear expression of the variables. Strategy-proofness and individual rationality can also easily be expressed as linear inequalities involving the variables. The optimal auction for discrete bids is derived via a mixed-integer program. For 2 bidders, the AMD solution takes only seconds. (This is completely expected. When the underlying problem has a polynomial-time solution, state-of-the-art MIP solvers often manage to solve it fast.) AMD does not scale to 3 or more bidders. The FPTAS is also restricted to 2 bidders. As mentioned in Papadimitriou & Pierrakos (2011), optimal auction design for correlated bidders becomes NP-hard to approximate when $n \geq 3$. For the above reasoning, AMD suffices as an alternative implementation.

MYERSON: We calculate every bidder's ironed virtual valuations based on their *marginal* distributions, instead of their *conditional* distributions. We then allocate the item to the bidder with the highest ironed virtual valuation. The item is not allocated if the highest ironed virtual valuation is below $0$. This auction guarantees monotonicity and therefore strategy-proofess, as all it does is to pretend that correlation does not exist and runs the optimal auction for independent bidders. The revenue of this auction is expected to be sub-optimal as it ignores correlation completely.

2ND: Second price auction with the optimal reserve price. The optimal reserve is calculated by simply trying all multiples of $0.001$.

## A.5 HAND-CRAFTED GRID DISTRIBUTIONS BASED ON MAGIC SQUARES

One baseline auction described in A.4 is MYERSON, which is Myerson's original optimal auction extended to correlated bidders, simply ignoring correlation altogether. This is achieved by pretending that the distribution is independent and using the marginal distribution of a bidder to calculate her ironed virtual valuation. To assess the amount of revenue lost by ignoring correlation, we resort to the *magic squares*. These are 2D matrices whose every row and every column sums up to the same value. Under grid distributions based on magic squares, every bidder's marginal distribution is always $U(0, 1)$. The following listed magic squares are from a 1531 book titled *De occulta philosophia*. We experimented on these magic-square-based grid distributions. The revenue gap between MYERSON (optimal if correlation does not exist) and our auction (proven near-optimal based on the upper bound) is the amount of revenue lost by ignoring correlation. For i.i.d. distributions, MYERSON is just the second price auction with a reserve based on the i.i.d. distributions.

$$\text{SATURN} = \begin{bmatrix} 4 & 9 & 2 \\ 3 & 5 & 7 \\ 8 & 1 & 6 \end{bmatrix}$$

$$\text{JUPITER} = \begin{bmatrix} 4 & 14 & 15 & 1 \\ 9 & 7 & 6 & 12 \\ 5 & 11 & 10 & 8 \\ 16 & 2 & 3 & 13 \end{bmatrix}$$

$$\text{MARS} = \begin{bmatrix} 11 & 24 & 7 & 20 & 3 \\ 4 & 12 & 25 & 8 & 16 \\ 17 & 5 & 13 & 21 & 9 \\ 10 & 18 & 1 & 14 & 22 \\ 23 & 6 & 19 & 2 & 15 \end{bmatrix}$$

$$\text{SOL} = \begin{bmatrix} 6 & 32 & 3 & 34 & 35 & 1 \\ 7 & 11 & 27 & 28 & 8 & 30 \\ 19 & 14 & 16 & 15 & 23 & 24 \\ 18 & 20 & 22 & 21 & 17 & 13 \\ 25 & 29 & 10 & 9 & 26 & 12 \\ 36 & 5 & 33 & 4 & 2 & 31 \end{bmatrix}$$

$$\text{VENUS} = \begin{bmatrix} 22 & 47 & 16 & 41 & 10 & 35 & 4 \\ 5 & 23 & 48 & 17 & 42 & 11 & 29 \\ 30 & 6 & 24 & 49 & 18 & 36 & 12 \\ 13 & 31 & 7 & 25 & 43 & 19 & 37 \\ 38 & 14 & 32 & 1 & 26 & 44 & 20 \\ 21 & 39 & 8 & 33 & 2 & 27 & 45 \\ 46 & 15 & 40 & 9 & 34 & 3 & 28 \end{bmatrix}$$

$$\text{MERCURY} = \begin{bmatrix} 8 & 58 & 59 & 5 & 4 & 62 & 63 & 1 \\ 49 & 15 & 14 & 52 & 53 & 11 & 10 & 56 \\ 41 & 23 & 22 & 44 & 45 & 19 & 18 & 48 \\ 32 & 34 & 35 & 29 & 28 & 38 & 39 & 25 \\ 40 & 26 & 27 & 37 & 36 & 30 & 31 & 33 \\ 17 & 47 & 46 & 20 & 21 & 43 & 42 & 24 \\ 9 & 55 & 54 & 12 & 13 & 51 & 50 & 16 \\ 64 & 2 & 3 & 61 & 60 & 6 & 7 & 57 \end{bmatrix}$$

$$\text{LUNA} = \begin{bmatrix} 37 & 78 & 29 & 70 & 21 & 62 & 13 & 54 & 5 \\ 6 & 38 & 79 & 30 & 71 & 22 & 63 & 14 & 46 \\ 47 & 7 & 39 & 80 & 31 & 72 & 23 & 55 & 15 \\ 16 & 48 & 8 & 40 & 81 & 32 & 64 & 24 & 56 \\ 57 & 17 & 49 & 9 & 41 & 73 & 33 & 65 & 25 \\ 26 & 58 & 18 & 50 & 1 & 42 & 74 & 34 & 66 \\ 67 & 27 & 59 & 10 & 51 & 2 & 43 & 75 & 35 \\ 36 & 68 & 19 & 60 & 11 & 52 & 3 & 44 & 76 \\ 77 & 28 & 69 & 20 & 61 & 12 & 53 & 4 & 45 \end{bmatrix}$$

## A.6 ADVERSARIAL DISTRIBUTIONS

After experimenting on $47$ distributions, including $10$ randomly generated grid distributions for $n$ from 2 to 5, and 7 hand-crafted grid distributions based on the magic squares in A.5, an unexpected observation is that for all these distributions, a near-optimal verifiably monotone allocation can be found without incorporating counterexample-guided training. The post-processing monotonicity fix step is therefore also not needed. For 2 bidders, we always find a verifiably monotone allocation with one trial. For more bidders, we sometimes do end up with non-monotone allocations, but this can be resolved by repeated trials. We do not need an excessively large number of trials. We used 10 trials in experiments. In the worst situation (5 bidders, distribution $G5_6$), we ended up with a verifiably monotone allocation in 6 out of 10 trials. That is, in expectation, we only need less than 2 trials to obtain a monotone allocation. Furthermore, for all these distributions, we achieve near-optimal revenue.

While the above observation shows the effectiveness of our approach, a natural question to ask is whether these results are artifacts caused by our selection of distributions. To test the limit of our approach, we experimented with two methods for generating adversarial distributions. The first method is based on evolutionary computation. We managed to generate distributions under which the greedy allocation is (probably) "wrong" for more bid profiles, in the sense that its decision is inconsistent with decisions obtained via Automated Mechanism Design. Nevertheless, our approach still delivers near-optimal results with one trial, without any counterexample-guided training. The evolutionary computation based adversarial distributions did increase the revenue gap between the greedy upper bound and the achieved revenue, from maximally $1.3\%$ for non-adversarial distributions to $2.1\%$. The second method relies on a mixed-integer program to generate adversarial distributions under which the greedy allocation maximally "overestimates" the revenue. This method successfully leads to two adversarial distributions for which we do need counterexample-guided training and monotonicity fix. That is, simply relying on luck (i.e., train again until we reach a verifiably monotone allocation with near-optimal revenue) no longer works. We do need the full suite of techniques (verification, counterexample-guided training and post-processing monotonicity fix) to achieve near-optimal revenue.

### A.6.1 ADVERSARIAL DISTRIBUTIONS VIA EVOLUTIONARY COMPUTATION

Our approach aims to find a monotone interpolation of the greedy allocation. One adversarial situation is when the learning target, i.e., the greedy allocation, makes allocation mistakes for many bid profiles. In this section, we apply evolutionary computation to search for a distribution under which the greedy allocation differs the most from the "correct" allocation (i.e., the optimal allocation).

However, the correct allocation is unknown. We settle for the AMD allocation instead, which we expect to be mostly correct. In A.4, we described the automated mechanism design (AMD) approach. The main gist of AMD is that we can round the bids down to multiples of $1/H$. Afterwards, the optimal auction for these discrete bids can be derived via mixed-integer programming. Since AMD is only scalable for 2 bidders, we focus on 2 bidders. Going through the $H^2$ bid profiles, we count the number of times greedy and AMD are inconsistent. This counter is used as the fitness function for evolutionary computation – we want its value to go up. In our experiment, we set $H$ to 20 to speed up the fitness evaluation.

We apply the simple (1+1)EA algorithm. We start with the grid distribution $G5_s$ for $s$ from 0 to 9. We use $d$ to denote the current distribution. In every evolutionary round, for every density value in the 2D matrix representing $d$, we replace the original density value $x$ by $0.9 \cdot x + u$, where $u$ is drawn from $U[0, 1]$. After mutation, the density values are normalized to ensure that the total probability equals 1. We evaluate the fitness of the mutated distribution $d'$. If it is better (i.e., more adversarial), then we keep it by replacing $d$ by $d'$. Otherwise, we throw $d'$ away and go back to $d$. We do the above for 100 rounds. The resulting distribution is recorded as $EA(G5_s)$.

Despite the effort, under the most adversarial distribution we found via evolutionary computation, greedy and AMD only differ under $3.5\%$ of the bid profiles. This empirically suggests that the greedy allocation is a high quality learning target even for (somewhat) adversarial distributions, justifying our approach. For the evolved adversarial distributions, our neural network interpolation approach still produces near-optimal revenue as shown in Table 3. Among all 59 distributions we experimented on, $EA(G5_2)$ leads to the worst revenue gap between the greedy upper bound and the

achieved revenue, which is 2.1%. (For non-adversarial distributions, the maximum revenue gap is 1.3%.)

Table 3: 2 bidders, evolutionary computation generated adversarial distributions

| DISTRIB. | GREEDY | AMD | MYER. | 2ND | RNET | MP |
|---|---|---|---|---|---|---|
| $EA(G5_0)$ | 0.3935 $\pm 0.001$ | 0.3918 | 0.3857 $\pm 0.0007$ | 0.3869 $\pm 0.0008$ | 0.3875 $\pm 0.0008$ | **0.3905** $\pm 0.0008$ |
| $EA(G5_1)$ | 0.4143 $\pm 0.001$ | 0.4111 | 0.4030 $\pm 0.0008$ | 0.4010 $\pm 0.0008$ | 0.4028 $\pm 0.0008$ | **0.4106** $\pm 0.0008$ |
| $EA(G5_2)$ | 0.4311 $\pm 0.001$ | 0.4279 | 0.4137 $\pm 0.0007$ | 0.4116 $\pm 0.0007$ | 0.3836 $\pm 0.001$ | **0.4220** $\pm 0.0008$ |
| $EA(G5_3)$ | 0.4165 $\pm 0.001$ | 0.4140 | 0.3999 $\pm 0.0008$ | 0.3986 $\pm 0.0008$ | 0.4033 $\pm 0.0008$ | **0.4117** $\pm 0.0008$ |
| $EA(G5_4)$ | 0.4271 $\pm 0.001$ | 0.4228 | 0.4152 $\pm 0.0007$ | 0.4156 $\pm 0.0008$ | 0.4175 $\pm 0.0008$ | **0.4260**$^*$ $\pm 0.0008$ |
| $EA(G5_5)$ | 0.4374 $\pm 0.001$ | 0.4320 | 0.4277 $\pm 0.0007$ | 0.4246 $\pm 0.0008$ | 0.4258 $\pm 0.0008$ | **0.4320**$^*$ $\pm 0.0007$ |
| $EA(G5_6)$ | 0.4381 $\pm 0.001$ | 0.4362 | 0.4256 $\pm 0.0008$ | 0.4279 $\pm 0.0008$ | 0.4307 $\pm 0.0008$ | **0.4336** $\pm 0.0008$ |
| $EA(G5_7)$ | 0.4025 $\pm 0.001$ | 0.3998 | 0.3912 $\pm 0.0007$ | 0.3934 $\pm 0.0008$ | 0.3958 $\pm 0.0008$ | **0.4002**$^*$ $\pm 0.0008$ |
| $EA(G5_8)$ | 0.4355 $\pm 0.001$ | 0.4320 | 0.4211 $\pm 0.0008$ | 0.4224 $\pm 0.0008$ | 0.4199 $\pm 0.0008$ | **0.4324**$^*$ $\pm 0.0008$ |
| $EA(G5_9)$ | 0.4141 $\pm 0.001$ | 0.4107 | 0.4068 $\pm 0.0008$ | 0.4076 $\pm 0.0008$ | 0.4051 $\pm 0.0008$ | **0.4097** $\pm 0.0009$ |

## A.6.2 ADVERSARIAL DISTRIBUTIONS VIA MIXED-INTEGER-PROGRAMMING

We explore another heuristic direction for finding adversarial distributions. We use $R^G$ to denote the greedy revenue, which is generally not attainable as greedy is often not strategy-proof. $R^G$ serves as the revenue upper bound. We use $R^*$ to denote the optimal revenue. We aim to find the distribution that maximizes the ratio $\frac{R^G}{R^*}$. That is, we are searching for distributions under which the greedy allocation significantly overestimates the revenue, which would make greedy an unsuitable learning target.

We still work within the structure of the grid distributions, focusing on $5 \times 5$ matrices as illustrated below. Our goal is to search for a set of 25 values for the $p_{i,j}$, so that the corresponding grid distribution is adversarial based on the metric mentioned above.

$$\text{WORST}(\lambda) = \begin{bmatrix} p_{0,4} & p_{1,4} & p_{2,4} & p_{3,4} & p_{4,4} \\ p_{0,3} & p_{1,3} & p_{2,3} & p_{3,3} & p_{4,3} \\ p_{0,2} & p_{1,2} & p_{2,2} & p_{3,2} & p_{4,2} \\ p_{0,1} & p_{1,1} & p_{2,1} & p_{3,1} & p_{4,1} \\ p_{0,0} & p_{1,0} & p_{2,0} & p_{3,0} & p_{4,0} \end{bmatrix}$$

We construct a mixed-integer program to find these 25 values. There are 25 continuous variables, i.e., the $p_{i,j}$, ranging from 1 to $\lambda$, where $\lambda$ is a distribution parameter. We derived two distributions called WORST10 (setting $\lambda = 10$) and WORST100 (setting $\lambda = 100$).

We face two challenges when constructing the mixed-integer program:

The first challenge is that the marginal profit is not linear in the probability densities. To make it linear, we re-interpret the above matrix as a discrete distribution. Every bidder's bid can only be from the following 5 values: $\{0, 0.2, 0.4, 0.6, 0.8\}$. $p_{0,0}$ refers to the probability that both bidders bid 0s. $p_{1,3}$ refers to the probability that bidder 1 bids 0.2 and bidder 2 bids 0.6. Essentially, for the original 2D grid distribution, there are 25 uniform sub-squares, we assign all probability mass to the bottom left corner of the sub-square to make the distribution discrete. By pretending that the distribution is discrete, the marginal profits become linear in the $p_{i,j}$ with the help of axillary binary variables. We use $r_{i,j}^k$ with $k \in \{1, 2\}$ and $0 \leq i, j \leq 4$ to represent the maximum revenue we can

extract from bidder $k$ when bidder 1 bids $i/5$ and bidder 2 bids $j/5$. We have

$$r^1_{i,j} = \max_{i' \geq i} \left( \frac{i'}{5} \sum_{i'' \geq i'} p_{i'',j} \right)$$

$$r^2_{i,j} = \max_{j' \geq j} \left( \frac{j'}{5} \sum_{j'' \geq j'} p_{i,j''} \right)$$

The marginal profit of bidder $k$ when bidder 1 bids $i/5$ and bidder 2 bids $j/5$ is denoted as $m^k_{i,j}$, which equals the following. (This is not the "per density" version. Please refer to A.2.)

$$m^1_{i,j} = \max\{r^1_{i,j} - r^1_{i+1,j}, 0\}$$

$$m^2_{i,j} = \max\{r^2_{i,j} - r^2_{i,j+1}, 0\}$$

The greedy revenue $R^G$ is then

$$\sum_{0 \leq i,j \leq 4} \max\{m^1_{i,j}, m^2_{i,j}\}$$

The second challenge is that the optimal revenue is unknown. We settle for an alternative revenue. We define two auctions for discrete bids. Auction 1 allocates the item to bidder 1 if and only if $b_1 \geq b_2$. Auction 2 allocates the item to bidder 1 if and only if $b_1 > b_2$. The only difference between these two auctions is the way they perform tie-breaking. Note that tie-breaking is consequential for discrete distributions. We evaluate the revenue of these two auctions using marginal profit. The revenue under auction 1 is

$$\sum_{0 \leq j \leq i \leq 4} m^1_{i,j} + \sum_{0 \leq i < j \leq 4} m^2_{i,j}$$

The revenue under auction 2 is

$$\sum_{0 \leq j < i \leq 4} m^1_{i,j} + \sum_{0 \leq i \leq j \leq 4} m^2_{i,j}$$

The maximum revenue between these two auctions is denoted as $R^S$.

Since the revenue of $R^S$ is evaluated in terms of marginal profit, the revenue fix is already automatically included. The revenue fix basically adds a reserve for each bidder and the reserve can depend on the other bid. The optimal reserve is already reflected in the marginal profits.

So far, all values mentioned above can be expressed as linear expressions of the $p_{i,j}$ (with the help of many auxiliary binary variables to represent "max"). We finally add a constraint $R^G \geq \alpha \cdot R^S$, where $\alpha$ is a constant. We search for the largest constant $\alpha$ that still makes the above inequality holds, which can be solved via a mixed-integer program (i.e., via feasibility check). The $p_{i,j}$ values corresponding to the largest $\alpha$ characterize the adversarial distribution. Below are the solutions for $\lambda \in \{10, 100\}$:

$$\text{WORST10} = \begin{bmatrix} 1 & 1 & 1 & 1 & 1 \\ 1 & 10 & 9 & 1 & 10 \\ 1 & 1 & 1 & 1 & 1 \\ 1 & 1 & 1 & 2.59 & 1 \\ 1 & 1 & 1 & 7.77 & 1 \end{bmatrix}$$

$$\text{WORST100} = \begin{bmatrix} 1 & 1 & 1 & 1 & 1 \\ 1 & 100 & 1 & 1 & 1 \\ 1 & 1 & 48.5 & 24.25 & 72.75 \\ 1 & 14.25 & 1 & 1 & 3 \\ 1 & 42.75 & 1 & 1 & 1 \end{bmatrix}$$

For WORST10 and WORST100, our approach still manages to produce near-optimal revenue, as shown in Table 4. However, these two distributions indeed are significantly more challenging to handle. Unlike other distributions where we can completely skip counterexample-guided training and monotonicity fix (by resorting to repeated trials), for WORST10 and WORST100, we do need the full suite of techniques. More detailed case studies on these two distributions are presented in A.7.

Table 4: 2 bidders, mixed-integer-programming generated adversarial distributions

| DISTRIB. | GREEDY | AMD | MYER. | 2ND | RNET | MP |
|---|---|---|---|---|---|---|
| WORST10 | 0.4401 | 0.4379 | 0.3733 | 0.3919 | 0.4130 | **0.4441**$^*$ |
| | ±0.001 | | ±0.0009 | ±0.0006 | ±0.0007 | ±0.003 |
| WORST100 | 0.4678 | 0.4565 | 0.3885 | 0.3805 | 0.4126 | **0.4632**$^*$ |
| | ±0.001 | | ±0.0007 | ±0.0004 | ±0.0006 | ±0.003 |

## A.7 CASE STUDY ON THE BENEFIT OF COUNTEREXAMPLE-GUIDED TRAINING AND MONOTONICITY FIX

To test the necessity and effectiveness of counterexample-guided training and monotonicity fix, we run the following experiment. For each selected distribution, we run 100 training trials to reach statistically meaningful conclusions. Each training trial proceeds as follows:

- Train $20,000$ (optimizer) steps without counterexample-guided training. We call the result at the end of $20,000$ steps the *stage-one* result.

- We verify whether the stage-one allocation is monotone and evaluate its revenue. If the stage-one allocation is already monotone, then we do not need to apply monotonicity fix when evaluating its revenue. Otherwise, the monotonicity fix is used.

- We continue to train another $10,000$ steps. If the stage-one allocation is not monotone, then we switch to counterexample-guided training and use the monotonicity violations from stage-one's evaluation as counterexamples. If the stage-one allocation is already monotone, then we do not need to switch to counterexample-guided training (and we do not have counterexamples anyway). We call the end result the *stage-two* result.

- We verify whether the stage-two allocation is monotone and evaluate its revenue. Similar to the situation for stage-one, we apply monotonicity fix when necessary.

The trials were divided into four categories, which are represented using different colours in Figure 3, Figure 4 and Figure 5:

- Blue: Stage-one and two are both monotone. That is, blue results do not involve counterexample-guided training or monotonicity fix.

- Red: Stage-one and two are both not monotone. Counterexample-guided training and monotonicity fix are both used.

- Green: Stage-one is not monotone and stage-two is monotone. That is, counterexample-guided training managed to guide the allocation to monotonicity. Both counterexample-guided training and monotonicity fix are used.

- Black: Stage-one is monotone but stage-two is not monotone. That is, with additional training, the network learned more details and turned not monotone. Monotonicity fix is used but not counterexample-guided training.

We perform a detailed analysis of the following 3 distributions. As mentioned earlier, in this experiment, we run 100 trails for each distribution. For 3 distributions, we estimate that the total computation time is on the order of $500$ hours, so we cannot afford to test every distribution.

- $G5_6$ for 5 bidders: This distribution is selected because it is more difficult to reach monotonicity under $G5_6$ compared to the other randomly generated distributions. When constructing Table 8, Table 9 and Table 10, for every distribution we tried 10 times and recorded how many times the result is monotone. For $G5_6$, 6 out of 10 times are monotone. $6/10$ appears to be a fairly high success rate, but is actually the worst among the randomly generated distributions.

- WORST10 and WORST100 for 2 bidders: These are adversarial distributions designed to test the limit of our approach. They are derived in A.6.2.

The experimental results are summarized in Table 5. The individual distribution's reports are given in Figure 3, Figure 4 and Figure 5.

Table 5: Summary of best revenue per category, where 100 trials are divided into 4 categories based on the monotonicity status at the end of stage-one and two: $M$=Monotonicity; $\overline{M}$=non-monotonicity; $M \to \overline{M}$ means the stage-one result is monotone and the stage-two result is not monotone

| DISTRIB. | BLUE $M \to M$ | RED $\overline{M} \to \overline{M}$ | GREEN $\overline{M} \to M$ | BLACK $M \to \overline{M}$ |
|---|---|---|---|---|
| $G5_6$ ($n = 5$) | $0.6841 \pm 0.0006$ | $0.6551 \pm 0.002$ | $0.6847 \pm 0.002$ | $0.6844 \pm 0.002$ |
| WORST10 ($n = 2$) | $0.4347 \pm 0.0009$ | $0.4441 \pm 0.003$ | $0.4410 \pm 0.003$ | $0.4370 \pm 0.003$ |
| WORST100 ($n = 2$) | $0.4517 \pm 0.0008$ | $0.4632 \pm 0.003$ | $0.4516 \pm 0.002$ | $0.4544 \pm 0.0008$ |

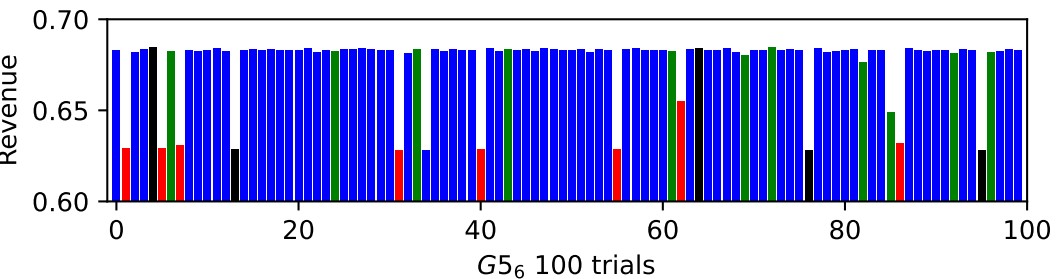

Figure 3: While green and black reached slightly higher highs, the gap between blue and green/black is statistically insignificant considering the standard errors. In conclusion, counterexample-guided training and monotonicity fix are not needed for this distribution.

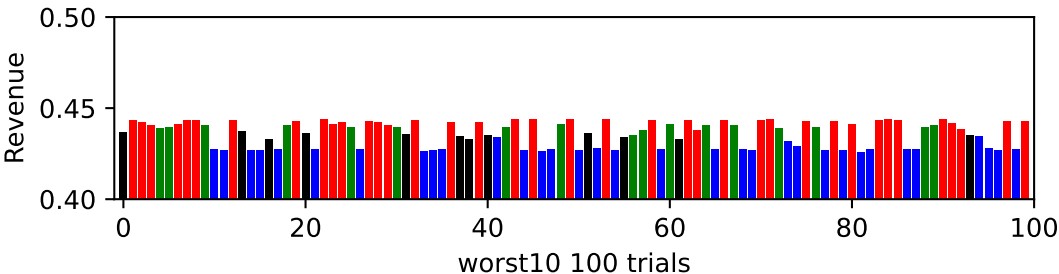

Figure 4: Red and green reached higher, so counterexample-guided training and monotonicity fix do help. We do need the whole suite of techniques to reach near-optimal revenue.

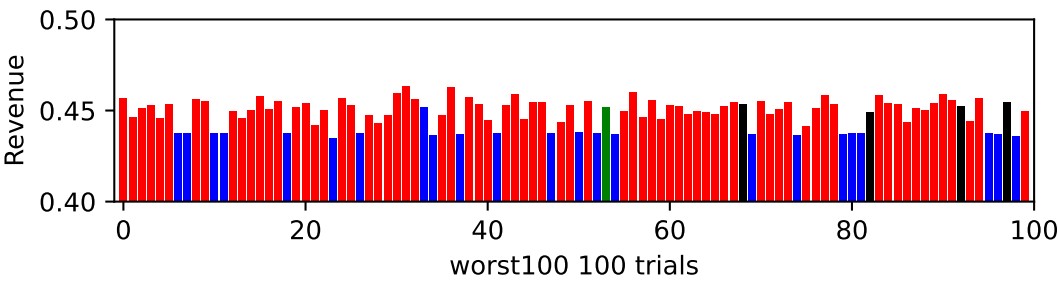

Figure 5: Red reached higher, so counterexample-guided training and monotonicity fix do help. We do need the whole suite of techniques to reach near-optimal revenue.

## A.8   DETAILS OF REGRETNET IMPLEMENTATION

RegretNet (Dütting et al., 2019) implements two neural networks, one for allocation and one for payments. In our RegretNet implementation, the two networks have the following properties:

1. Each network takes the $n$ bids as inputs.

2. Each network is a multilayer perceptron (MLP) with ReLU activation. In our implementation, both networks have 2 intermediate layers each with width 50. This is larger than the networks used in the main approach, which is 2 layers with width 20.

3. The allocation network has $n + 1$ outputs where the first $n$ outputs correspond to the $n$ bidders and the last output corresponds to no allocation. We take the argmax among the $n + 1$ outputs to decide which bidder wins. In training, we take softmax on the outputs.

4. The payment network has $n$ outputs where the $i$-th output decides bidder $i$'s payment if the item is allocated to bidder $i$.

RegretNet is unsupervised. The training samples are bid profiles drawn according to the given correlated distribution. Unlike the main approach, where we need the distribution, as otherwise we cannot calculate the marginal profits, RegretNet does not need the access to the distribution. Samples are all it needs.

Following RegretNet's notation, we define the set of bidders as $N$, denote the model parameters as $w$, denote $v_i$ as the true valuation of bidder $i$. Bidder $i$'s utility is $u_i^w(v_i; (v_i', v_{-i}))$ when she has valuation $v_i$ and bids $v_i'$. Bidder $i$'s payment is denoted as $p_i^w(v)$.

The aim of RegretNet is to maximize the expected total payment $\mathbb{E}[\sum_{i \in N} p_i^w(v)]$ while ensuring strategy-proofness as well as ex post individual rationality. During training, we use a batch of $L$ bid profiles to get an unbiased estimation $\frac{1}{L} \sum_{l=1}^{L} \sum_{i \in N} p_i^{w,l}(v)$.

The strategy-proofness constraint is given by minimizing the "Regret" defined as the sum of how profitable the optimal deviation is for every bidder. Formally,

$$rgt_i(w) = \mathbb{E} \left[ \max_{v_i' \in V_i} u_i^w(v_i; (v_i', v_{-i})) - u_i^w(v_i; (v_i, v_{-i})) \right].$$

During training, we use a batch of bid profiles to give an unbiased estimation $\widehat{rgt}_i(w)$.

Individual rationality is ensured by taking the sigmoid of the payment network's outputs, so that the range becomes $[0, 1]$, then times the bid. This method can ensure that the payment from each bidder is nonnegative and no more than the bid itself, thus satisfying individual rationality.

The loss function used in unsupervised training is as follows:

$$Loss(w; \lambda) = -\frac{1}{L} \sum_{l=1}^{L} \sum_{i \in N} p_i^{w,l}(v) + \sum_{i \in N} \lambda_i \widehat{rgt}_i(w).$$

In the original implementation of RegretNet (Dütting et al., 2019), there is also a second order term, which we dropped. The factor $\lambda_i$'s are initialized to 50 and updated periodically. We set the update interval to be 20 backward steps. The original RegretNet uses a gradient descent method to find the optimal update direction and amount, while in our implementation, we set it higher if $\widehat{rgt}_i(w)$ is larger than 0.0001, and smaller if it is less than it. 0.0001 is the largest regret that we tolerate.

Since the allocation network is using the same network architecture as the main approach, i.e., MLP+ReLU, the existing suite of MIP-based techniques still apply, including verification, counterexample-guided training, and monotonicity fix. After training, the payment network is thrown away and we only evaluate the allocation network. The existing evaluation process from the main approach still applies. For all results on RegretNet listed in this paper, the auction is verified 100% strategy-proof (no counterexample-guided training was needed). For the distributions in Table 7, our RegretNet implementation is able to outperform MYERSON and 2ND, and achieve revenue that is reasonably close to our main approach, noting again that our main approach requires more than just the samples, but also the exact distribution. RegretNet also outperforms MYERSON and 2ND for 8 out of 12 adversarial distributions (Table 3 and Table 4). On the other hand, for the distributions in Table 8, Table 9 and Table 10 (i.e., $n = 3, 4, 5$), the revenue achieved is less ideal. We set the same training budget for our main approach and RegretNet, which is $20,000$ optimizer steps. For 3 or more bidders, perhaps RegretNet requires a lot more training resources. After all, it trains two networks instead of one and the networks are significantly larger. We did not perform excessive hyper-parameter tuning for the RegretNet approach as RegretNet is not the main focus of this paper (mostly used as a baseline). Furthermore, given its larger size, it is a lot more time consuming to work with. For example, for the 10 auctions in Table 10, MIP-based verification takes 21 to 100 minutes.

## A.9 Alternative network architectures

The concept of monotonicity represented by the min-max networks (Sill, 1997) is different from the concept of allocation monotonicity in our mechanism design context. For example, in our model, the $i$-th input of the allocation network is agent $i$'s bid and the $i$-th output of the allocation network represents the priority value of agent $i$. Whoever has the highest priority value wins. Allocation monotonicity basically requires the following: if the $i$-th output is currently the highest, then when we increase the $i$-th input, we want the $i$-th output to still be the highest. The $i$-th output does not have to be monotone with respect to the $i$-th input – it is fine that an agent's priority value drops when she increases her bid, as long as she still wins (for example, the other agents' priority values may drop more, or the winner's priority value drop is not big enough to go below another agent). It is not clear how to reconcile the difference between monotonicity in terms of values between inputs and outputs and allocation monotonicity. In other words, it is not clear how to use the min-max networks to represent the whole space of strategy-proof allocations.

There are two ways to use min-max networks to represent allocation functions in our setting. The first is MyersonNet (Dütting et al., 2019). Under MyersonNet, agent $i$'s bid $b_i$ is mapped to $i$'s allocation priority value $f_i(b_i)$, where $f_i$ is nondecreasing function represented by a min-max network. Whoever has the highest priority value wins. This representation is strategy-proof, but the following example shows that MyersonNet cannot represent the full space of strategy-proof mechanisms and sometimes it leads to significant revenue loss. In our experiments, we used the fully strategy-proof variant of RegretNet (Dütting et al., 2019) as a baseline, which is more expressive than MyersonNet.

*Example:* We have two bidders. Bidder 2's value is either $0$ or $\epsilon$ (an infinitesimal value), *i.e.*, almost all revenue will come from bidder 1. When bidder 2's value is $0$, bidder 1's conditional distribution is $D_0$. When bidder 2's value is $\epsilon$, bidder 1's conditional distribution is $D_1$. That is, bidder 2 serves the purpose of sending a signal that tells us whether bidder 1's distribution is $D_0$ or $D_1$. Since almost all revenue comes from bidder 1, a near-optimal auction has the following form: When bidder 2's value is $0$, the auction becomes a take-it-or-leave-it auction for bidder 1 with an optimal reserve price derived based on $D_0$, which we call RESERVE($D_0$). When bidder 2's value is $\epsilon$, the auction becomes a take-it-or-leave-it auction for bidder 1 with an optimal reserve price derived on $D_1$, which we call RESERVE($D_1$). We further assume that RESERVE($D_0$) > RESERVE($D_1$). MyersonNet cannot express the above auction. Under MyersonNet, when bidder 2's bid increases, bidder 1's winning price never drops. Basically, under MyersonNet, bidder 1 must face sub-optimal reserve under either $D_0$ or $D_1$ (or both), and it is easy to construct distributions where sub-optimal reserves significantly impact the revenue.

Roughgarden & Talgam-Cohen (2013) mentioned the *Lopomo assumption*. Using our terminology, if a bidder's allocation priority value is increasing in her own bid and decreasing in others' bids, then allocation monotonicity is easily satisfied. We can certainly use the min-max networks to model such kind of allocations, but once again, the auction in the above example cannot be expressed. In the above example near-optimal auction, bidder 1, without changing her own bid, can go from losing to winning when bidder 2 increases her bid, which goes against the Lopomo assumption.

## A.10 Multi-unit auctions with unit demand

We extend our techniques to multi-unit auctions with unit demand. We use $m$ to denote the number of items, where $m < n$. Myerson's greedy allocation is extended by assigning one item to each bidder with the $m$ highest marginal profit.

In the single-item setting, our neural network's output dimension is $n+1$. If the $i$-th ($i = 1, 2, \ldots, n$) output coordinate is the highest, then bidder $i$ wins. If the $(n + 1)$-th coordinate is the highest, then the item is not allocated. For the multi-unit setting, the network's output dimension remains $n + 1$. Bidder $i$ wins if and only if the $i$-th output coordinate is among the $m$ highest output coordinates and the $i$-th output coordinate is higher than the $(n+1)$-th output coordinate. This allows us to represent all possible numbers of items allocated (*i.e.*, from 0 to $m$).

The extended monotonicity condition requires that any winner $i$, who is in the original *winner set*, must remains in the winner set when $i$ increases her bid while the other bids stay the same. This more complex version requires us to enumerate all combinations of $i$, all original winner sets that contain $i$ and all new winner sets that exclude $i$. The monotonicity verification MIP for the single-item setting can be trivially adapted, as any given winner set can be described by a set of inequalities.

Below we present our experimental results. The baseline is the $(m + 1)$-th price auction with the optimal reserve price. For each distribution, we run 10 trials, and in every trial, we stop after $5,000$, $10,000$, $15,000$ and $20,000$ optimizer steps to perform monotonicity verification. We record the best-performing verifiably monotone allocations.

Table 6: Multi-unit auctions with unit demand with 2 items (i.e., $m = 2$)
M+1 refers to the (m+1)-th price auction with the optimal reserve price
$\text{GAP}(\text{MP}) = 0.67\%$, $\text{GAP}(\text{M+1}) = 5.1\%$

| Distrib. | $n = 3$ | | | $n = 4$ | | | $n = 5$ | | |
|---|---|---|---|---|---|---|---|---|---|
| | Greedy | M+1 | MP | Greedy | M+1 | MP | Greedy | M+1 | MP |
| $G5_0$ | 0.7629 | 0.7251 | **0.7624** | 0.9387 | 0.8923 | **0.9360** | 1.0856 | 1.0428 | **1.0757** |
| | ±0.002 | ±0.001 | ±0.001 | ±0.002 | ±0.001 | ±0.001 | ±0.002 | ±0.001 | ±0.001 |
| $G5_1$ | 0.7896 | 0.7413 | **0.7901** | 0.9646 | 0.9172 | **0.9608** | 1.0973 | 1.0477 | **1.0844** |
| | ±0.002 | ±0.001 | ±0.001 | ±0.002 | ±0.001 | ±0.001 | ±0.002 | ±0.001 | ±0.001 |
| $G5_2$ | 0.7311 | 0.6900 | **0.7311** | 0.9380 | 0.8932 | **0.9319** | 1.0868 | 1.0394 | **1.0755** |
| | ±0.002 | ±0.001 | ±0.001 | ±0.002 | ±0.001 | ±0.001 | ±0.002 | ±0.001 | ±0.001 |
| $G5_3$ | 0.7497 | 0.7048 | **0.7500** | 0.9344 | 0.8853 | **0.9293** | 1.0930 | 1.0462 | **1.0799** |
| | ±0.002 | ±0.001 | ±0.001 | ±0.002 | ±0.001 | ±0.001 | ±0.002 | ±0.001 | ±0.001 |
| $G5_4$ | 0.7724 | 0.7280 | **0.7733** | 0.9457 | 0.8959 | **0.9397** | 1.0900 | 1.0456 | **1.0737** |
| | ±0.002 | ±0.001 | ±0.001 | ±0.002 | ±0.001 | ±0.001 | ±0.002 | ±0.001 | ±0.001 |
| $G5_5$ | 0.7511 | 0.6956 | **0.7473** | 0.9329 | 0.8841 | **0.9266** | 1.0938 | 1.0463 | **1.0748** |
| | ±0.002 | ±0.001 | ±0.001 | ±0.002 | ±0.001 | ±0.001 | ±0.002 | ±0.001 | ±0.001 |
| $G5_6$ | 0.7400 | 0.6911 | **0.7401** | 0.9507 | 0.8964 | **0.9449** | 1.0965 | 1.0474 | **1.0852** |
| | ±0.002 | ±0.001 | ±0.001 | ±0.002 | ±0.001 | ±0.001 | ±0.002 | ±0.001 | ±0.001 |
| $G5_7$ | 0.8122 | 0.7734 | **0.8096** | 0.9663 | 0.9201 | **0.9643** | 1.0945 | 1.0498 | **1.0825** |
| | ±0.002 | ±0.001 | ±0.001 | ±0.002 | ±0.001 | ±0.001 | ±0.002 | ±0.001 | ±0.001 |
| $G5_8$ | 0.7531 | 0.6924 | **0.7525** | 0.9390 | 0.8936 | **0.9349** | 1.0925 | 1.0462 | **1.0785** |
| | ±0.002 | ±0.001 | ±0.001 | ±0.002 | ±0.001 | ±0.001 | ±0.002 | ±0.001 | ±0.001 |
| $G5_9$ | 0.7949 | 0.7329 | **0.7934** | 0.9682 | 0.9130 | **0.9604** | 1.1016 | 1.0523 | **1.0894** |
| | ±0.002 | ±0.001 | ±0.001 | ±0.002 | ±0.001 | ±0.001 | ±0.002 | ±0.001 | ±0.001 |

A.11 COMPLETE TABLES INCLUDING STANDARD ERRORS

Table 7: 2 bidders (with standard errors)
$\text{GAP(MP)} = 0.26\%, \text{GAP(RNET)} = 2.7\%, \text{GAP(2ND)} = 4.2\%, \text{GAP(MYER)} = 4.3\%$

| DISTRIB. | GREEDY | AMD | MYER. | 2ND | RNET | MP | VV | IVV |
|---|---|---|---|---|---|---|---|---|
| $G5_0$ | 0.4353 ±0.001 | 0.4368 | 0.4191 ±0.0008 | 0.4197 ±0.0009 | 0.4307 ±0.0009 | **0.4368** ±0.0009 | 0.4355 ±0.0009 | 0.4329 ±0.0009 |
| $G5_1$ | 0.4047 ±0.001 | 0.4041 | 0.3939 ±0.0007 | 0.3929 ±0.0006 | 0.4021 ±0.0007 | **0.4037** ±0.0007 | 0.4032 ±0.0007 | 0.4032 ±0.0007 |
| $G5_2$ | 0.3979 ±0.001 | 0.3967 | 0.3771 ±0.0009 | 0.3823 ±0.001 | 0.3869 ±0.0009 | **0.3959** ±0.001 | 0.3946 ±0.001 | 0.3955 ±0.001 |
| $G5_3$ | 0.4629 ±0.001 | 0.4625 | 0.4533 ±0.0006 | 0.4546 ±0.0006 | 0.4522 ±0.0007 | **0.4628**$^*$ ±0.0006 | 0.4612 ±0.0007 | 0.4616 ±0.0007 |
| $G5_4$ | 0.4681 ±0.001 | 0.4665 | 0.4476 ±0.0009 | 0.4472 ±0.0009 | 0.4639 ±0.0008 | **0.4674**$^*$ ±0.0008 | 0.4673 ±0.0008 | 0.4671 ±0.0008 |
| $G5_5$ | 0.4204 ±0.001 | 0.4187 | 0.4075 ±0.0008 | 0.4057 ±0.0008 | 0.4062 ±0.0009 | **0.4151** ±0.0009 | 0.4148 ±0.001 | 0.4150 ±0.001 |
| $G5_6$ | 0.3905 ±0.001 | 0.3912 | 0.3761 ±0.0007 | 0.3741 ±0.0008 | 0.3798 ±0.0008 | **0.3908** ±0.0007 | 0.3893 ±0.0008 | 0.3886 ±0.0008 |
| $G5_7$ | 0.4865 ±0.001 | 0.4854 | 0.4494 ±0.0006 | 0.4544 ±0.0008 | 0.4771 ±0.0007 | **0.4850** ±0.0008 | 0.4828 ±0.0008 | 0.4840 ±0.0008 |
| $G5_8$ | 0.4298 ±0.001 | 0.4273 | 0.3988 ±0.001 | 0.3981 ±0.0009 | 0.4077 ±0.0009 | **0.4291**$^*$ ±0.0008 | 0.4281 ±0.0009 | 0.4279 ±0.0009 |
| $G5_9$ | 0.4531 ±0.001 | 0.4512 | 0.4306 ±0.0009 | 0.4302 ±0.0009 | 0.4379 ±0.0009 | **0.4498** ±0.0009 | 0.4482 ±0.001 | 0.4497 ±0.0009 |
| SATURN | 0.4533 ±0.001 | 0.4545 | 0.4210 ±0.0008 | 0.4226 ±0.0008 | 0.4372 ±0.0008 | 0.4510 ±0.0008 | 0.4512 ±0.0008 | **0.4515** ±0.0008 |
| JUPITER | 0.4427 ±0.001 | 0.4404 | 0.4238 ±0.0008 | 0.4219 ±0.0009 | 0.4316 ±0.0008 | **0.4418**$^*$ ±0.0009 | 0.4409 ±0.0009 | 0.4410 ±0.0009 |
| MARS | 0.4375 ±0.001 | 0.4377 | 0.4177 ±0.0008 | 0.4167 ±0.0008 | 0.4194 ±0.0008 | **0.4360** ±0.0009 | 0.4333 ±0.0009 | 0.4359 ±0.0009 |
| SOL | 0.4429 ±0.001 | 0.4414 | 0.4376 ±0.0008 | 0.4364 ±0.0008 | 0.4329 ±0.0009 | **0.4427**$^*$ ±0.0008 | 0.4404 ±0.0008 | 0.4419 ±0.0008 |
| VENUS | 0.4381 ±0.001 | 0.4381 | 0.4175 ±0.0008 | 0.4171 ±0.0008 | 0.4233 ±0.0008 | **0.4365** ±0.0008 | 0.4319 ±0.0009 | 0.4358 ±0.0008 |
| MERCURY | 0.4273 ±0.001 | 0.4274 | 0.4179 ±0.0008 | 0.4176 ±0.0008 | 0.4150 ±0.0008 | **0.4281**$^*$ ±0.0008 | 0.4271 ±0.0008 | 0.4279 ±0.0008 |
| LUNA | 0.4331 ±0.001 | 0.4335 | 0.4173 ±0.0008 | 0.4172 ±0.0008 | 0.4212 ±0.0008 | **0.4322** ±0.0008 | 0.4315 ±0.0009 | 0.4318 ±0.0009 |

Table 8: 3 bidders (with standard errors)
$\text{GAP(MP)} = 0.36\%, \text{GAP(RNET)} = 3.6\%, \text{GAP(2ND)} = 3.9\%, \text{GAP(MYER)} = 3.8\%$

| DISTRIB. | GREEDY | MYER. | 2ND | RNET | MP |
|---|---|---|---|---|---|
| $G5_0$ | 0.5512±0.001 | 0.5346±0.0007 | 0.5338±0.0008 | 0.5346±0.0007 | **0.5517**±0.0007 |
| $G5_1$ | 0.5647±0.0009 | 0.5467±0.0007 | 0.5475±0.0007 | 0.5509±0.0007 | **0.5651**±0.0007 |
| $G5_2$ | 0.5316±0.001 | 0.5137±0.0007 | 0.5118±0.0008 | 0.5153±0.0008 | **0.5302**±0.0008 |
| $G5_3$ | 0.5476±0.001 | 0.5283±0.0007 | 0.5282±0.0007 | 0.5290±0.0007 | **0.5450**±0.0007 |
| $G5_4$ | 0.5485±0.0009 | 0.5297±0.0007 | 0.5306±0.0007 | 0.5310±0.0007 | **0.5471**±0.0007 |
| $G5_5$ | 0.5477±0.001 | 0.5237±0.0007 | 0.5231±0.0007 | 0.5255±0.0007 | **0.5443**±0.0008 |
| $G5_6$ | 0.5371±0.001 | 0.5155±0.0008 | 0.5138±0.0008 | 0.5158±0.0008 | **0.5332**±0.0007 |
| $G5_7$ | 0.5723±0.0009 | 0.5527±0.0007 | 0.5528±0.0007 | 0.5555±0.0006 | **0.5710**±0.0007 |
| $G5_8$ | 0.5473±0.001 | 0.5186±0.0008 | 0.5193±0.0007 | 0.5185±0.0007 | **0.5429**±0.0008 |
| $G5_9$ | 0.5663±0.001 | 0.5409±0.0007 | 0.5410±0.0008 | 0.5396±0.0007 | **0.5637**±0.0008 |

Table 9: 4 bidders (with standard errors)

$\text{GAP(MP)} = 0.78\%$, $\text{GAP(RNET)} = 3.5\%$, $\text{GAP(2ND)} = 3.2\%$, $\text{GAP(MYER)} = 3.2\%$

| DISTRIB. | GREEDY | MYER. | 2ND | RNET | MP |
|---|---|---|---|---|---|
| $G5_0$ | $0.6270\pm0.0009$ | $0.6086\pm0.0007$ | $0.6082\pm0.0007$ | $0.6072\pm0.0006$ | $\mathbf{0.6242}\pm0.0007$ |
| $G5_1$ | $0.6417\pm0.0009$ | $0.6215\pm0.0006$ | $0.6205\pm0.0007$ | $0.6192\pm0.0006$ | $\mathbf{0.6379}\pm0.0007$ |
| $G5_2$ | $0.6250\pm0.0009$ | $0.6096\pm0.0007$ | $0.6101\pm0.0007$ | $0.6072\pm0.0006$ | $\mathbf{0.6205}\pm0.0007$ |
| $G5_3$ | $0.6266\pm0.0009$ | $0.6054\pm0.0006$ | $0.6055\pm0.0006$ | $0.6034\pm0.0006$ | $\mathbf{0.6220}\pm0.0007$ |
| $G5_4$ | $0.6296\pm0.0009$ | $0.6084\pm0.0006$ | $0.6077\pm0.0007$ | $0.6076\pm0.0006$ | $\mathbf{0.6233}\pm0.0007$ |
| $G5_5$ | $0.6213\pm0.0009$ | $0.6029\pm0.0007$ | $0.6028\pm0.0007$ | $0.6020\pm0.0006$ | $\mathbf{0.6158}\pm0.0007$ |
| $G5_6$ | $0.6347\pm0.0009$ | $0.6099\pm0.0007$ | $0.6085\pm0.0007$ | $0.6051\pm0.0006$ | $\mathbf{0.6287}\pm0.0007$ |
| $G5_7$ | $0.6427\pm0.0009$ | $0.6210\pm0.0006$ | $0.6234\pm0.0006$ | $0.6193\pm0.0006$ | $\mathbf{0.6369}\pm0.0006$ |
| $G5_8$ | $0.6286\pm0.0009$ | $0.6100\pm0.0007$ | $0.6091\pm0.0007$ | $0.6085\pm0.0007$ | $\mathbf{0.6241}\pm0.0007$ |
| $G5_9$ | $0.6411\pm0.0009$ | $0.6199\pm0.0006$ | $0.6201\pm0.0006$ | $0.6190\pm0.0006$ | $\mathbf{0.6357}\pm0.0007$ |

Table 10: 5 bidders (with standard errors)

$\text{GAP(MP)} = 0.87\%$, $\text{GAP(RNET)} = 2.8\%$, $\text{GAP(2ND)} = 2.4\%$, $\text{GAP(MYER)} = 2.4\%$

| DISTRIB. | GREEDY | MYER. | 2ND | RNET | MP |
|---|---|---|---|---|---|
| $G5_0$ | $0.6862\pm0.0008$ | $0.6695\pm0.0006$ | $0.6700\pm0.0006$ | $0.6670\pm0.0006$ | $\mathbf{0.6799}\pm0.0006$ |
| $G5_1$ | $0.6901\pm0.0008$ | $0.6737\pm0.0006$ | $0.6738\pm0.0006$ | $0.6713\pm0.0005$ | $\mathbf{0.6846}\pm0.0006$ |
| $G5_2$ | $0.6838\pm0.0008$ | $0.6709\pm0.0006$ | $0.6691\pm0.0006$ | $0.6681\pm0.0005$ | $\mathbf{0.6783}\pm0.0006$ |
| $G5_3$ | $0.6894\pm0.0008$ | $0.6724\pm0.0006$ | $0.6721\pm0.0006$ | $0.6684\pm0.0005$ | $\mathbf{0.6838}\pm0.0006$ |
| $G5_4$ | $0.6853\pm0.0008$ | $0.6692\pm0.0006$ | $0.6682\pm0.0006$ | $0.6664\pm0.0006$ | $\mathbf{0.6787}\pm0.0006$ |
| $G5_5$ | $0.6850\pm0.0008$ | $0.6717\pm0.0006$ | $0.6720\pm0.0006$ | $0.6685\pm0.0006$ | $\mathbf{0.6809}\pm0.0006$ |
| $G5_6$ | $0.6902\pm0.0008$ | $0.6710\pm0.0006$ | $0.6725\pm0.0006$ | $0.6686\pm0.0006$ | $\mathbf{0.6839}\pm0.0006$ |
| $G5_7$ | $0.6900\pm0.0008$ | $0.6719\pm0.0006$ | $0.6715\pm0.0006$ | $0.6682\pm0.0005$ | $\mathbf{0.6824}\pm0.0006$ |
| $G5_8$ | $0.6868\pm0.0008$ | $0.6707\pm0.0006$ | $0.6715\pm0.0006$ | $0.6659\pm0.0006$ | $\mathbf{0.6807}\pm0.0006$ |
| $G5_9$ | $0.6927\pm0.0008$ | $0.6752\pm0.0006$ | $0.6768\pm0.0006$ | $0.6711\pm0.0005$ | $\mathbf{0.6866}\pm0.0006$ |

## A.12   ADDITIONAL EXPERIMENTS FOR ICLR REBUTTAL

### A.12.1   EXPERIMENTS ON MONOTONE NETWORKS

We present experiments on four additional approaches:

- MYERNET (Dütting et al., 2019): Agent $i$'s bid $b_i$ is mapped to $f_i(b_i)$, where $f_i$ is a nondecreasing function. $f_i(b_i)$ represents agent $i$'s allocation priority value. The agent with the highest priority value wins the item. If all agents' priority values are below 0, then the item is thrown away. The monotonicity of the $f_i$ guarantees strategy-proofness. The $f_i$ are represented using the monotone min-max networks (Sill, 1997). We train the $f_i$ using the standard RegretNet (Dütting et al., 2019) approach. We simultaneously train $n + 1$ networks. Each $f_i$ is a min-max network with 5 "max" groups and each group contains 5 nodes. There is also a shared payment network, which is a MLP with 2 hidden layers and 20 nodes for each layer. (Despite the tiny networks, in practise, it is already fairly expensive to train, with more details given below.) The payment network is only used during training. After training, the payment network is thrown away and the "correct" payments are derived from the (architecturally guaranteed) monotonic allocation function. All other implementation details follow our RegretNet implementation described in A.8.

- MINMAX: This is a generalised version of MYERNET. Agent $i$'s priority value $f_i(b_i, b_{-i})$ now also depends on the others' bids. $f_i$ is nondecreasing in $b_i$ and nonincreasing in every dimension in $b_{-i}$. This representation also guarantees strategy-proofness. We also use the min-max networks to represent the $f_i$.[10] Same as MYERNET, each $f_i$ is a min-max network with 5 "max" groups and each group contains 5 nodes. The only difference is that the input dimension of $f_i$ is now $n$ instead of 1. All other settings are the same as MYERNET.

*On training difficulty:* Both MYERNET and MINMAX are fairly expensive to train. For experiments reported in this subsection, every instance takes about 12 hours on average to train. We conducted our experiments (20 instances) in parallel on a high-performance cluster with Intel 8360Y CPUs. As mentioned earlier, in our experiments, every min-max network contains only 5 groups and each group contains 5 nodes. If we enlarge the network size to 10 groups and 10 nodes each group, then training becomes too expensive. Based on our estimation, *each* instance takes 30+ hours even with an Nvidia A100 GPU. (Unfortunately, at the moment, we do not have access to multiple GPUs.)

Below we present two modified approaches that are significantly more scalable, by adopting the neural network interpolation idea from our paper.

- MYERNET+: We apply our paper's main approach to improve MYERNET. That is, we apply supervised training to train only the allocation function (i.e., the $f_i$). The supervision goal is to train the $f_i$ to replicate exactly Myerson's greedy allocation – the agent with the highest (conditional) virtual valuation should win the item and the item is thrown away if all virtual valuations are below 0.

  Each $f_i$ is represented using a min-max network with 10 "max" groups and each group contains 10 nodes. All other implementation details follow our main approach described in A.3. Each training instance takes around 35 minutes with an Intel 8360Y CPU.

  There are several reasons why MYERNET+ is significantly faster: 1) there is no need to calculate regret, which is the most time consuming step; 2) we switch from unsupervised training to supervised; 3) we have one less network to train as the payment network is no longer needed.

- MINMAX+: Same as MYERNET+, we apply our paper's main approach to improve MIN-MAX. Same as the case for MYERNET+, the supervision goal is to train the $f_i$ to replicate exactly Myerson's greedy allocation. Each $f_i$ is also represented using a min-max network with 10 "max" groups and each group contains 10 nodes. Each training instance also takes around 35 minutes with an Intel 8360Y CPU.

---

[10]By flipping the sign of input dimension $i$, we can change the output from being nondecreasing to nonincreasing in dimension $i$.

*On structural limitation of* MYERNET *and* MINMAX: In A.9, we described an example showing that MYERNET/MINMAX style allocations may lead to significant revenue loss. Here we further elaborate on that example using specific numbers.

Admittedly, the following is a fairly contrived example, but it is presented to illustrate the limitation of MYERNET and MINMAX. We assume that the bid profile is either $(1, 0)$ or $(0.5, \epsilon)$, each with 50% chance. That is, with 50% chance, bidder 1's value is 1 and bidder 2's value is 0, and with 50% chance, bidder 1's value is 0.5 and bidder 2's value is $\epsilon$, where $\epsilon$ is infinitesimal. A near optimal mechanism works as follows: if bidder 2's value is 0, then bidder 1 faces a take-it-or-leave-it offer of 1, and if bidder 2's value is $\epsilon$, then bidder 1 faces a take-it-or-leave-it offer of 0.5. The expected revenue is 0.75. Suppose our allocation follows the style of either MYERNET or MINMAX. We use $p_0$ to represent bidder 1's critical price for winning when bidder 2 bids 0 and we use $p_\epsilon$ to represent bidder 1's critical price for winning when bidder 2 bids $\epsilon$. Since we assume our allocation follows the style of either MYERNET or MINMAX, we must have $p_0 \le p_\epsilon$. For example, suppose the allocation follows the style of MINMAX, then when bidder 2's bid increases from 0 to $\epsilon$, bidder 1's priority value either stays the same or drops and bidder 2's priority value either stays the same or increases. In order for bidder 1 to still beat bidder 2, bidder 1's minimum winning bid must either stay the same or increase. If $p_\epsilon \le 0.5$, then the maximum revenue extracted from bidder 1 is then at most $0.5p_0 + 0.5p_\epsilon \le 0.5$. If $p_\epsilon > 0.5$, then the maximum revenue extracted from bidder 1 is then at most $0.5p_0 \le 0.5$. Therefore, the maximum revenue extracted from both bidders is at most $0.5 + \epsilon$. Earlier, we showed that the optimal revenue is at least 0.75. That is, by adopting MYERNET or MINMAX, for this example, we lose one third of the revenue.

Below we present experimental results on randomly generated grid distributions ($G5_0$ to $G5_9$) for 5 bidders. We still use the revenue gap (to the unattainable greedy upper bound) as the performance indicator. Our method MP's revenue gap is 0.87%. The revenue gaps of MYERNET+ and MIN-MAX+, both are based on techniques proposed in our paper, are 3.3% and 3.4%, respectively. For reference, we have also included the revenues of the manual baseline MYER., which is Myerson's greedy allocation where virtual valuations are calculated based on the *marginal* distributions (essentially ignoring correlation altogether). Both MYERNET+ and MINMAX+ do not outperform MYER. Lastly, MYERNET and MINMAX have much worse performances with gaps at 19% and 18%.

Table 11: Additional experiments for 5 bidders
GAP(MP) = 0.87%, GAP(MYER.) = 2.4%
GAP(MYERNET+) = 3.3%, GAP(MYERNET) = 19%
GAP(MINMAX+) = 3.4%, GAP(MINMAX) = 18%

| DISTRIB. | GREEDY | MYERNET | MINMAX | MYERNET+ | MINMAX+ | MYER. | MP |
|---|---|---|---|---|---|---|---|
| $G5_0$ | 0.6862 | 0.5754 | 0.5135 | 0.6630 | 0.6625 | 0.6695 | **0.6799** |
|  | ±0.0008 | ±0.0005 | ±0.0008 | ±0.0006 | ±0.0006 | ±0.0006 | ±0.0006 |
| $G5_1$ | 0.6901 | 0.5746 | 0.4503 | 0.6681 | 0.6670 | 0.6737 | **0.6846** |
|  | ±0.0008 | ±0.0005 | ±0.0007 | ±0.0006 | ±0.0006 | ±0.0006 | ±0.0006 |
| $G5_2$ | 0.6838 | 0.5216 | 0.5688 | 0.6642 | 0.6635 | 0.6709 | **0.6783** |
|  | ±0.0008 | ±0.0006 | ±0.0006 | ±0.0006 | ±0.0006 | ±0.0006 | ±0.0006 |
| $G5_3$ | 0.6894 | 0.6549 | 0.4469 | 0.6665 | 0.6659 | 0.6724 | **0.6838** |
|  | ±0.0008 | ±0.0006 | ±0.0009 | ±0.0006 | ±0.0006 | ±0.0006 | ±0.0006 |
| $G5_4$ | 0.6853 | 0.5809 | 0.6270 | 0.6632 | 0.6627 | 0.6692 | **0.6787** |
|  | ±0.0008 | ±0.0006 | ±0.0005 | ±0.0006 | ±0.0006 | ±0.0006 | ±0.0006 |
| $G5_5$ | 0.6850 | 0.6051 | 0.5904 | 0.6650 | 0.6642 | 0.6717 | **0.6809** |
|  | ±0.0008 | ±0.0005 | ±0.0006 | ±0.0006 | ±0.0006 | ±0.0006 | ±0.0006 |
| $G5_6$ | 0.6902 | 0.6139 | 0.6369 | 0.6661 | 0.6658 | 0.6710 | **0.6839** |
|  | ±0.0008 | ±0.0006 | ±0.0005 | ±0.0006 | ±0.0006 | ±0.0006 | ±0.0006 |
| $G5_7$ | 0.6900 | 0.6041 | 0.6161 | 0.6659 | 0.6655 | 0.6719 | **0.6824** |
|  | ±0.0008 | ±0.0006 | ±0.0006 | ±0.0006 | ±0.0006 | ±0.0006 | ±0.0006 |
| $G5_8$ | 0.6868 | 0.2196 | 0.6210 | 0.6644 | 0.6638 | 0.6707 | **0.6807** |
|  | ±0.0008 | ±0.0008 | ±0.0006 | ±0.0006 | ±0.0006 | ±0.0006 | ±0.0006 |
| $G5_9$ | 0.6927 | 0.6312 | 0.5969 | 0.6688 | 0.6680 | 0.6752 | **0.6866** |
|  | ±0.0008 | ±0.0005 | ±0.0006 | ±0.0006 | ±0.0006 | ±0.0006 | ±0.0006 |

### A.12.2 CONTINUOUS DISTRIBUTIONS

Lastly, we present a few additional experiments on continuous distributions. We experimented with three continuous distributions for 2 bidders. We use $f(x, y)$ to denote the joint probability density function.[11] Our method MP is near optimal. On the other hand, 2ND (second price auction with optimal reserve) also performs quite well for all three cases.

Table 12: Experiments on three continuous distributions
$\mathrm{GAP}(\mathrm{MP}) = 0.02\%, \mathrm{GAP}(2\mathrm{ND}) = 0.5\%$

| DISTRIB. | GREEDY | 2ND | MP |
|---|---|---|---|
| $f(x, y) = 2x + y$ | $0.5095 \pm 0.001$ | $0.5079 \pm 0.0008$ | $\mathbf{0.5094} \pm 0.0008$ |
| $f(x, y) = 2x - y + 1$ | $0.4518 \pm 0.001$ | $0.4469 \pm 0.0008$ | $\mathbf{0.4517} \pm 0.0008$ |
| $f(x, y) = sin(x) + cos(y) + 2$ | $0.4216 \pm 0.001$ | $0.4211 \pm 0.0008$ | $\mathbf{0.4215} \pm 0.0008$ |

---

[11] We need to apply a normalisation factor $\alpha$ to ensure that $\int_0^1 \int_0^1 \alpha f(x, y) dx dy = 1$. We ignore $\alpha$ for presentation purpose.

