# OpenReview forum: "Extending Myerson's Optimal Auctions to Correlated Bidders via Neural Network Interpolation"
_ICLR.cc/2025/Conference — Submitted to ICLR 2025_

### Official Review · Reviewer_oxsU · 2024-11-03

**Soundness:** 2
**Presentation:** 1
**Contribution:** 2
**Rating:** 5
**Confidence:** 3

**Summary:**

In single-item sealed-bid auctions, Myerson characterized the revenue-maximizing mechanism under the assumption that the value distributions of all bidders are independent. However, this result does not hold when bidders' values are correlated. In this paper, the authors propose learning a parameterized auction mechanism from sampled data to maximize revenue while maintaining strategy-proofness.

**Strengths:**

1. The discussion of preliminary and related work is thorough.
2. The setting of correlated bidders is more general than that of classical independent bidders.
3. The authors conduct experiments across a wide range of valuation distributions.

**Weaknesses:**

1. The presentation could be further improved. The authors allocate excessive space to the Introduction and preliminaries (Sections 2 and 3), leaving less room for the methodology (Section 4), which I believe is the core of the paper.
2. The methodology is unclear to me. It appears that the authors use a neural network to compute the randomized allocation results. However, how is the neural network trained? Do you set the parameters using Mixed Integer Programming (MIP)?
3. If the authors trained the neural network using MIP, how is the monotonicity condition satisfied? If there are any violations, these should be noted and included in the experimental results, similar to what was done in previous literature RegretNet.

**Questions:**

See "Weaknesses".

---

> ### Author Response · Authors · 2024-11-16
>
> We thank the reviewer for the comments!
>
> The reviewer mentioned that relative to introduction/preliminaries, we allocated less space for methodology in the main text.  We deferred a lot of technical discussion as well as the implementation details to the appendix, which has 18 pages.  We will consider trimming down the introduction and move some of the technical details back to the main text.  Regarding presentation quality, we want to point out that reviewer rnaD rated our presentation as "excellent" and reviewer RXmb rated our presentation as "good".
>
> The training method is at the bottom of page 7. We do not use MIP for training. Training is simply gradient descent. The cost function is given at the bottom of page 7. The network architecture and hyperparameters are given in Appendix A.3. We are happy to clarify (and modify the future versions) if the reviewer thinks any part is not clear.
>
> All our mechanisms are 100% strategy-proof. On page 8, there are 2 MIPs. The MIP on the left is used for monotonicity verification. If the training result is fully monotone, then we have reached strategy-proofness. If the verification MIP failed, then we resort to repeated trials and counterexample guided training. If we still cannot achieve monotonicity, then we apply the MIP on the right, which post processes the allocation, which fixes any monotonicity violations.

---

### Official Review · Reviewer_rnaD · 2024-11-03

**Soundness:** 4
**Presentation:** 4
**Contribution:** 3
**Rating:** 6
**Confidence:** 4

**Summary:**

The paper is another in the recent thread of work on deep learning for revenue-maximizing auction design (part of what is sometimes called “differentiable economics”).

Here, the focus is on *single-parameter* auctions. Here, Myerson (1981) was an essential contribution, which gave closed-form solutions for the optimal strategyproof auction assuming independent bidders.

Myerson’s simple idea is to use information about the distribution of bidder types to convert each bidder’s valuation into a “virtual valuation”, and then allocate to maximize that virtual valuation. Myerson also showed that any strategyproof mechanism must have a monotone allocation rule. Under certain reasonable conditions, maximizing virtual valuation is monotone (and if these conditions are not met, virtual valuation can be “ironed”).

Myerson’s full result holds only when bidders are independent. When the bidders are correlated, maximizing virtual valuation (ironed or not) may not be monotone. So finding a revenue-maximizing auction is much harder.

The idea of this paper is to use neural networks to search for high-revenue allocation rules and then, by embedding the neural networks in integer programs, to certify that those allocation rules are monotone (therefore strategyproof). Since these certification programs can also generated counterexamples for monotonicity, those can be used in the training loop to gradually remove monotonicity violations.

The authors show that their approach works well on a number of correlated valuation distributions, and can even be extended to single-parameter multi-unit settings.

**Strengths:**

- Interesting approach for a reasonable mechanism design problem and some nice generalizations
- Approach works empirically well
- Pleasantly clear discussion of extensions of Myerson to correlated valuations, and the challenges that result
- Certification programs are of general interest (i.e. one can apply them to RegretNet too to certify strategyproofness)

**Weaknesses:**

- Single-parameter correlated bidder auction design is in some sense a relatively niche problem
- Neural networks used are necessarily tiny to be able to be certified. This has its positives but is a limitation if more representational capacity is required for a given problem.
- The authors make a case for their grid distributions, but it would be nice to see some more common correlated distributions from the literature, in particular those with continuous support. The neural network approaches should be able to handle this.

**Questions:**

- I’m a little bit confused by the discussion of monotonicity. I think I grasp it, but I didn’t find appendix A.9 all that helpful. It might be useful to clarify why the typical Rochet characterization doesn’t do all the work here, as there are obvious ways of making cyclically monotone allocation rules (e.g. take the gradient of an input-convex NN).
- See weaknesses: why not at least take a shot on continuous-support distributions?

---

> ### Author Response · Authors · 2024-11-16
>
> We thank the reviewer for the encouraging comments!
>
> We respectively disagree that our model is relatively niche.  Myerson's optimal auction is widely regarded as a foundational result in economic theory, often considered one of the most significant results in economic theory.  Its significance is underscored by its role in earning Roger Myerson the Nobel Prize in Economics in 2007.  Myerson's model is on selling one item to multiple independent bidders.  With correlation, the model becomes a long-standing hard problem.  Empirically, our techniques deliver excellent performance. We achieve near-optimal revenues on all distributions experimented and beat all existing baselines in terms of revenue gaps from the unattainable greedy upper bound, sometimes improving the best baseline by tenfold.
>
> The reviewer is correct that our approach is able to handle continuous distributions. (But it does take some effort as the existing code base is written only considering grid distributions.) We have uploaded a new version of our paper. We added a new subsection titled "Additional experiments for ICLR rebuttal" (the last 3 pages at the end of the appendix). In particular, we added experiments on three continuous joint probability density functions:
>
> a) f(x,y)=2x+y
>
> b) f(x,y)=2x-y+1
>
> c) f(x,y)=sin(x)+cos(x)+2
>
> (Note that the above probability density functions need to be normalised to ensure that the total probability is 1. The normalisation ratio is ignored for presentation purpose.)
>
> Our method is still near optimal with an average revenue gap of only 0.02%.  On the other hand, second price auction with optimal reserve also performs quite well for all three cases, with an average revenue gap of 0.5%.  Our impression is that continuous distributions are generally easy to handle.  In Appendix A.6.2, we described two adversarial grid distributions. Take Worst100 as an example, its probability density function involves "sudden jumps", i.e., the probability density changes by 100 times within a small region. We conjecture that such jumps are the actual difficult cases for our approach. The intuition is that small networks are generally only capable of interpolating smooth allocations (and we expect the allocations to be smooth for smooth distributions).
>
> By Rochet characterization, we assume that the reviewer's suggestion is to interpolate the utilities instead of the allocation. I think this is challenging for two reasons: 1) we focus on deterministic auctions, so we still need to verify that the utilities are feasible via deterministic auctions (i.e., no over allocation); 2) there is an easy way to convert allocations to revenue, which is the whole point of the virtual valuations, but it is not clear how to conveniently convert utilities to payments during training.  Our cost function is simply the dot product of the allocation vector and the virtual valuation vector.  We hope the above makes sense, but it is possible that we completely misunderstood the reviewer's suggestion, so we are happy to discuss this further! And we thank the reviewer on this suggestion.  We definitely agree that the Rochet characterization is a good direction to pursue. On a side note, we are already considering it for the digital good model -- there is no over allocation and good auctions need to be randomised.

---

> > ### Comment · Reviewer_rnaD · 2024-11-22
> > **Response to authors**
> >
> > Thanks for the interesting discussion of continuous distributions. While the results themselves are not so exciting, I think it's really valuable to include them.
> >
> > I see your point about overallocation (this is also the issue in IID multi-parameter auctions). If your auctions are strategyproof, they will obey the Rochet characterization, but I can now see that just enforcing that isn't sufficient for this problem, so the architecture has to be different.

---

### Official Review · Reviewer_RXmb · 2024-11-06

**Soundness:** 4
**Presentation:** 3
**Contribution:** 4
**Rating:** 8
**Confidence:** 4

**Summary:**

The paper tackles the optimal auction design in a two-bidder case with correlated valuation. The key challenge of naively adopting Myerson’s approach to this problem is that the resulting allocation rule is not guaranteed monotone due to the correlation between the value distributions. This problem was shown to be hard in general more than 10 years ago.

The paper proposes an empirical approach to this problem, where a neural network is trained to approximate the greedy allocation while subject to monotonicity regularization. This is done with the following components:
* Use a small and simple NN (MLP with ReLU) to encode the allocation function, and train it to maximize empirical revenue.
* A MIP (MIP-verify) is designed to verify the monotonicity of the allocation rule encoded by such an NN. MIP-verify in fact also (in some sense) quantifies the non-monotonicity of the NN, so it is also used as a monotonicity regularization during the training.
* Another MIP (MIP-fix) is designed to fix a non-monotone NN and generate a guaranteed monotone allocation rule, which is also the final output.

Empirical results suggest that the mechanisms obtained through this approach significantly outperform those obtained via traditional methods. In many test distributions, the gaps with respect to the unachievable greedy allocation benchmark get reduced by roughly 10x. The paper also shows that this approach can be generalized to multi-bidder cases but with unit demand settings.

**Strengths:**

This paper significantly improves the results on a fundamental auction design setting where no good enough theoretical results are known. The approach used nicely combined NN with MIP and guarantees provable strategy-proofness, which is usually difficult with empirical driven methods.

**Weaknesses:**

I don’t see any major weakness of this paper. One concrete suggestion is, whether it is possible to take one specific example distribution and present the corresponding greedy allocation and the allocation obtained from this method. It may help give more concrete intuitions for theorists and may inspire theoretical results as well.

**Questions:**

1. Why are there some rows in Table 1 where MP is larger than Greedy?
2. What are the major blockers for generalizing this method to general multi-bidder cases?

---

> ### Author Response · Authors · 2024-11-16
>
> We thank the reviewer for the encouraging comments!
>
> We will include visualizations of greedy and our allocation.
>
> The achieved revenue and the greedy upper bound are both calculated by averaging over 100000 samples. In the appendix, we include the full tables with standard errors.
>
> To generalise, we need to be able to formulate a MIP to verify strategy-proofness. If we bring the payment function back in, then it is generally doable as we can check for regret. If we only work with the allocation function, then it is doable if we are dealing with single-minded bidders (i.e., demand is a fixed bundle of heterogeneous items). Right now, it is not clear how to formulate such a MIP for general combinatorial auctions. I think I need more time to think this through.

---

### Official Review · Reviewer_EFY3 · 2024-11-07

**Soundness:** 3
**Presentation:** 2
**Contribution:** 2
**Rating:** 3
**Confidence:** 4

**Summary:**

In this paper, they authors propose a neural network based approach to automatically design revenue-optimal single item auctions with correlated valuation. The proposed approach includes (1) a post-processing to interpolate the greedy allocation to make the allocation monotone and (2) a revenue fix step to increase the revenue. The authors run many empirical analysis to verify the efficiency of their approach and show that it can be extended to multi-unit auction settings.

**Strengths:**

This paper analyzed a very interesting problem and lies in the intersection between machine learning and mechanism design. To understand the revenue-optimal single item auction with correlated valuation is important in practice, given it fits a lot of practical scenario.

The experiment results are sound and promising.

**Weaknesses:**

I indeed have some concerns in terms of the contribution of this paper.

1. I am not quite convinced about the contribution of the paper. The authors mentioned 4 main contributions in the introduction, however, it seems the techniques are a combination of several previous paper, (1) MIP using MLP with RELU activation is from [Curry et al. 2020]. The counter-example guided training is from [Guo 2024]. Can you clarify what is the main novelty of this paper? Is it just the usage of marginal profit in neural network?

2. The paper is not well-written. I think the authors put a lot of effort to highlight the contribution in words, but missed many details of the exact methodology used in the paper. For example, the "method" section (section 4) is fairly short and very condense. It seems requires a lot of prior knowledge to follow the techniques. When I read it, it seems everything seems already known in the previous literature. The only new is the new representation of NN using marginal profit and this leads to my first point.

3. I understand the theoretical limitation of the MyersonNet, which is basically relies on the independent assumption and each $f_i(b_i)$ is to learn the ironed virtual value function. However, I believe it is still valid to compare it empirical to show it doesn't work in practice.

4. Another concern is that the post-processing seems requires the knowledge of the joint distribution $\phi$, is it true? If we only have sample access to the distribution, can your approach still be applied?

Some minor comments:
Better to define the offer more explicitly, e.g., posted price or fixed price depends on the others' bids. Same comments also apply to other terminology.

**Questions:**

Besides my questions in the "Weaknesses", I have some questions as follows:

1. The second MIP is not always applied, right? If it passed the monotonicity check in the first MIP, you don't need the second MIP?

2. If I understand correctly, the problem of MyersonNet is due to the independent assumption, is it possible to generalize it as follows: There is a literature called lattice network, which can output a general monotonic network for multi-dimensional input. Therefore, we can have n lattice network for each bidder, where each lattice takes all bids as an input and the $i$th lattice network will make the output increasing with $b_i$ but decreasing with $b_{-i}$. In the final layer we can have a softmax layer of n lattice network to output an allocation probability. Therefore, the network can capture the correlated information across bids and still montone with each of dimension.  Lattice network is already implemented in tensorflow (https://www.tensorflow.org/lattice/overview?hl=en) so it should be easy to try. What do you think?

---

> ### Author Response · Authors · 2024-11-16
>
> We thank the reviewer for the comments!
>
> === on contributions of this paper ===
>
> We first summarize the main contributions of this paper:
>
> 1) A very common task in mechanism design is to verify a given manual mechanism's strategy-proofness. For example, in this paper, we want to check whether Myerson's optimal auction is still strategy-proof under a specific correlated distribution.
>
>     The above task is challenging especially when the mechanism's description is complex. For example, Myerson's optimal auction involves the calculation of convex hull (i.e., it involves the Graham scan algorithm from computational geometry).
>
>     We propose a general way to carry out strategy-proofness verification. We first use neural network to interpolate the mechanism. In this paper, we implement Myerson's optimal auction in code. (As long as the manual mechanism has a constructive description, we can always implement it in code.) This piece of code is then "translated" to a neural network (i.e., a bunch of weights and biases) via neural network interpolation. At this point, we can adopt the existing strategy-proofness verification technique by [Curry et al. 2020] (but customised to our model).
>
>     In summary, our novel contribution is the proposal that *neural network interpolation* can serve as a general tool for verifying mechanism properties, including but not limited to strategy-proofness. The mechanism may be expressed mathematically or in code.
>
> 2) We experimented with three monotonicity-seeking techniques:
>
>     a) repeated trials (i.e., if verification fails, train again)
>
>     b) counterexample-guided training
>
>     c) post-processing monotonicity fix
>
>     We conducted extensive experiments (based on hundreds of hours of computation) comparing these techniques. Please refer to Figure 3-5. Indeed, counterexample-guided training was first used in [Guo 2024] (it was not used for achieving strategy-proofness though). The other two techniques are new in this paper. We also argue that adopting the state-of-the-art technique in our paper is a good thing.
>
>     Other new techniques proposed in this paper include
>
>     d) integrating virtual valuation in training, which allows us to train the allocation function by itself, without involving the payment function. It is faster and easier to train only the allocation function, compared to training both allocation and payment functions simultaneously.
>
>     e) the revenue fix trick
>
>     In summary, technique (a),(c),(d),(e) are all novel.
>
> 3) We delivered excellent (i.e., near-optimal) empirical results for a fundamental mechanism design model, beating existing baselines and reducing the revenue gap to the unattainable greedy upper bound sometimes by tenfold.
>
> 4) We demonstrated generality of our technique by integrating our technique to RegretNet to design 100% strategy-proof mechanisms and also generalise to multi-unit auctions with unit demand.

---

> ### Author Response · Authors · 2024-11-16
>
> === MyersonNet and Lattice Network ===
>
> We have uploaded a new version of our paper. We added a new subsection titled "Additional experiments for ICLR rebuttal" (the last 3 pages at the end of the appendix). We added new experiments on four additional approaches:
>
> 1) MyerNet (i.e., MyersonNet): b_i is mapped to f_i(b_i), where f_i is a nondecreasing function, represented using the min-max network [Sill 97].  The agent with the highest f_i(b_i) value wins. The item is thrown away if all f_i(b_i) values are below 0.  We train it using the standard RegretNet approach. That is, training also involves a payment function.  The payment function is thrown away at the end of the training and the correct payments are derived from the (architecturally guaranteed) monotonic allocation function.
>
> 2) MinMax: This is a generalised version of MyersonNet. Agent i's priority value is f_i(b_i, b_{-i}). f_i is nondecreasing in b_i and nonincreasing in every dimension of b_{-i}. Since our code base is PyTorch, we did not use Lattice Network to achieve this. It is fairly easy to represent f_i also using the min-max network. (By flipping the sign of input dimension i, we can change the output from being nondecreasing to nonincreasing in dimension i.)
>
> Both MyerNet and MinMax are fairly expensive to train. For 5 bidders, every instance takes about 12 hours to train using an Intel 8360Y CPU. We conducted experiments on 20 instances in parallel on a high-performance cluster. (GPU-based training does not significantly improve the training speed based on our experiments even with an A100 GPU. Furthermore, we do not have access to multiple GPUs.)
>
> To speed up MyerNet and MinMax, we also tried two other approaches that are significantly more scalable, by adopting the neural network interpolation idea from our paper.
>
> 3) MyerNet+: We apply our paper's main approach to improve MyerNet. That is, we apply supervised training to train only the allocation function.  That is, the supervision goal is to train the f_i to replicate exactly Myerson's greedy allocation -- the agent with the highest (conditional) virtual valuation should win the item and the item is thrown away if all virtual valuations are below 0.
>
>     There are several reasons why MyerNet+ is significantly faster: 1) there is no need to calculate regret, which is the most time consuming step; 2) we switch from unsupervised training to supervised; 3) we have one less network to train as the payment network is no longer needed.
>
> 4) MinMax+: Same as MyerNet+, the supervision goal is to train the f_i (now depending on both b_i and b_{-i}) to replicate exactly Myerson's greedy allocation.
>
> For MyerNet+ and MinMax+, each training instance takes around 35 minutes with an Intel 8360Y CPU.
>
> We conducted experiments on 10 randomly generated grid distributions for 5 bidders, using seed 0 to 9.  Our method MP's average revenue gap (to the unattainable greedy upper bound) is 0.87%.  The revenue gap of MyerNet+ and MinMax+ are 3.3% and 3.4%. Actually, they perform worse than the manual baseline "Myer", which is Myerson's auction based on the *marginal* virtual valuations -- essentially ignoring correlation altogether.  Lastly, MyerNet and MinMax have much worse performances with gaps at 19% and 18%.
>
> === other comments ===
>
> Yes, our approach requires the knowledge of the joint probability density function. We could use samples to learn it first. If we have access to a learnt probability density function, then all our techniques still apply. Strategy-proofness still holds.
>
> We thank the reviewer for the comments on presentation. We will work on improving it for future versions.
>
> The reviewer is correct that if the verification MIP passes, then we do not need to run the post-processing monotonicity fix MIP.

---

### Meta-Review · Area_Chair_dkhm · 2024-12-20

**Metareview:**

This paper constructs an auction mechanism for single-item auctions, when the bidders value are correlated. Indeed, in that case, Myerson optimal auction is proved to be sub-optimal (at least not strategy-proof).

The contribution is purely empirical, it consists in creating a network architecture to mimic the original Myerson allocation while ensuring monotonicity (which is sufficient to ensure strategy-proofness).

Auctions with correlated valuations are notoriously difficult, so this contribution is a first step. However, there are not enough guarantees on the efficiency of this approach.

The reviewers also raised several concerns and questions that should be answered.

**Additional Comments On Reviewer Discussion:**

This paper had a large range of grade 3-5 and 6-8, so as many on the negative side than on the positive side, and thus it was borderline. Therefore, I went through it myself, and as I said above, I tend to agree with the (slightly) negative comments.

This is why I decided this borderline paper unfortunately did not make the cut.

---

### Decision · Program_Chairs · 2025-01-22

Reject